# Identifying the ubiquitination targets of E6AP by orthogonal ubiquitin transfer

Yiyang Wang [1], Xianpeng Liu[2], Li Zhou[1], Duc Duong[3], Karan Bhuripanyo[1,4], Bo Zhao[5], Han Zhou[1], Ruochuan Liu[1], Yingtao Bi[6], Hiroaki Kiyokawa[2] & Jun Yin[1]

E3 ubiquitin (UB) ligases are the ending modules of the E1–E2-E3 cascades that transfer UB to cellular proteins and regulate their biological functions. Identifying the substrates of an E3 holds the key to elucidate its role in cell regulation. Here, we construct an orthogonal UB transfer (OUT) cascade to identify the substrates of E6AP, a HECT E3 also known as Ube3a that is implicated in cancer and neurodevelopmental disorders. We use yeast cell surface display to engineer E6AP to exclusively transfer an affinity-tagged UB variant (xUB) to its substrate proteins. Proteomic identification of xUB-conjugated proteins in HEK293 cells affords 130 potential E6AP targets. Among them, we verify that MAPK1, CDK1, CDK4, PRMT5, β-catenin, and UbxD8 are directly ubiquitinated by E6AP in vitro and in the cell. Our work establishes OUT as an efficient platform to profile E3 substrates and reveal the cellular circuits mediated by the E3 enzymes.

[1] Department of Chemistry, Center for Diagnostics & Therapeutics, Georgia State University, Atlanta, GA 30303, USA. [2] Department of Pharmacology, Northwestern University, Chicago, IL 60611, USA. [3] Integrated Proteomics Core, Emory University, Atlanta, GA 30322, USA. [4] Department of Chemistry, University of Chicago, Chicago, IL 60637, USA. [5] Engineering Research Center of Cell & Therapeutic Antibody, Ministry of Education, and School of Pharmacy, Shanghai Jiao Tong University, Shanghai, 200240, China. [6] Department of Preventive Medicine, Northwestern University, Chicago, IL 60611, USA. Correspondence and requests for materials should be addressed to H.K. (email: kiyokawa@northwestern.edu) or to J.Y. (email: junyin@gsu.edu)

Ubiquitin (UB), a 76-residue protein riding on a E1–E2–E3 enzymatic cascade, is a key messenger in cell signaling[1]. UB attachment to cellular proteins regulates many key processes such as protein degradation, subcellular trafficking, enzymatic turnover, and complex formation. E1 activates UB with the formation of a thioester linkage between a catalytic Cys of E1 and the C-terminal Gly of UB[2]. UB bound to E1 is loaded on an E2 in a thioester exchange reaction to form a UB~E2 conjugate ("~" designates the thioester bond)[3]. E2 then carries UB to an E3 that recruits target proteins for UB conjugation[4–6]. The human genome encodes 2 E1s, at least 40 E2s and more than 600 E3s[3, 7, 8]. Since E3s recognize protein ubiquitination targets, they often play key regulatory roles, and their malfunction drives the development of many diseases including cancer, neurodegeneration, and inflammation[9, 10]. For example, E6AP, also known as Ube3a, is a E3 with a signature HECT domain for E2 binding[11]. E6AP is a critical regulator of neuron development; loss of its activity results in Angelman syndrome (AS), and duplications of chromosomal region 15q11-13 including its encoding gene *Ube3a* are associated with autism spectrum disorders (ASD)[12–15]. E6AP promotes tumorigenesis upon infection of high-risk human papillomavirus—it forms a complex with the viral oncoprotein E6 to ubiquitinate p53 and induce its degradation[11, 16]. Other non-HECT E3s may bind the E2~UB conjugate through a Ring, Ring-between-Ring (RBR) or U-box motif[4, 6, 7]. Regardless of the type of interactions with E2s, an E3 may uptake UB from multiple E2s, and various E3s transfer UB to an overlapping pool of substrates. The complex cross-reactivities among E2, E3, and substrates make it a significant challenge to profile the substrates of a specific E3 to map it on the cell signaling network.

We envision an "orthogonal UB transfer (OUT)" pathway in which a UB variant (xUB) is confined to a single track of engineered xE1, xE2, and xE3 would guide the transfer of xUB exclusively to the substrate of a specific E3 ("x" designates engineered UB or enzyme variants orthogonal to their native partners)[17]. By expressing xUB and the OUT cascade of xE1–xE2–xE3 in the cell and purifying cellular proteins conjugated to xUB, we would be able to identify the direct substrates of an E3. The development of the OUT cascade removes the cross-reacting paths among various E2s and E3s. It enables the assignment of E3 substrates by directly following xUB transfer through the E3 instead of reading some indirect indicators of protein ubiquitination such as affinity binding with E3, or change of protein stability or ubiquitination levels upon E3 expression.

To implement OUT, we need to engineer orthogonal pairs of xUB–xE1, xE1–xE2, and xE2–xE3 that are free of cross-reactivities with native E1, E2, and E3 to secure the exclusive transfer of xUB to the substrates of an E3 in the cell. We previously reported engineering orthogonal xUB–xE1 and xE1–xE2 pairs by phage display[17]. We also generated the xUB-xE1 pairs with the two human E1, Uba1, and Uba6, respectively, to differentiate their targets of UB transfer in the cell[18]. Here we report that we have accomplished the last leg of OUT engineering: we used yeast cell surface display to engineer an orthogonal xE2–xE3 pair with the HECT E3 E6AP; we expressed the OUT cascade in HEK293 cells to profile E6AP substrates; and we identified a number of key signaling proteins as E6AP substrates and established regulatory circuits mediated by UB transfer through E6AP.

## Results

### Constructing the xUB-xUba1 and the xUba1-xUbcH7 pair.
We previously generated an xUB-xE1 pair with the E1 enzyme Uba1 from *S. cerevisiae*[17]. Using phage selection, we found that the two mutations in xUB (R42E and R72E) would block xUB recognition by wt Uba1, yet by incorporating mutations Q576R, S589R and

D591R into the adenylation domain of yeast Uba1, we could restore the activity of xUB with E1 to form xUB~E1 thioester conjugates (Supplementary Fig. 1a, b). We also introduced mutations E1004K, D1014K and E1016K into the UFD domain of the yeast Uba1 to block its interaction with the wt E2s (Supplementary Fig. 1c). We then used phage display to identify mutations in the N-terminal helix of the Ubc1, a yeast E2, to restore E1–E2 interaction and enable UB transfer to the E2 enzyme. By combining the mutations in the adenylation and the UFD domains of yeast Uba1, we generated the E1 mutant xUba1 that can specifically transfer xUB to xUbc1, the E2 mutant from phage selection (Table 1). In contrast, xUB cannot be activated by wt Uba1 for its transfer to wt E2s. xUba1 cannot activate wt UB, neither can it transfer xUB to wt E2. Thus, the xUB–xE1 and xE1–xE2 pairs are orthogonal to their native partners, and they can assemble a two-step cascade to transfer xUB to a designated E2.

Our success in engineering the xUB-xE1 and xE1-xE2 pairs with the yeast system is instrumental for constructing the OUT cascade in the human cell. We found xUB is not catalytically active with either of the human E1s, Uba1 (also known as Ube1), or Uba6[18]. Since Uba1 plays a major role in supporting protein ubiquitination in the human cells[8], we decided to engineer human Uba1 as the xE1 for the OUT cascade. Based on the sequence homology between the human and yeast Uba1, we identified residues Q608, S621 and D623 in the adenylation domain and E1037, D1047, and E1049 in the UFD domain of human Uba1 matching the sites of mutations in the yeast Uba1 (Supplementary Fig. 1d, e). We mutated these residues to R or K of the opposite charge to generate human xUba1 (Table 1). As expected, we found human xUba1 is reactive with xUB by forming xUB~xUba1 conjugate, yet it rejects wt UB in the activation reaction (Fig. 1a). Moreover, xUba1 cannot transfer xUB to wt human E2s such as UbcH7 due to the mutations in the UFD domain of Uba1.

To restore xUB transfer to E2s, we generated orthogonal xE1-xE2 pairs based on the sequence homology between the yeast and human E2s. The N-terminal helix of E2 plays a key role in binding the UFD domain of the E1 as shown in the crystal structures of yeast *S. pombe* Uba1 in complex with E2 Ubc4, and the modeled structure of *S. cerevisiae* Uba1 bound with Ubc1 (Supplementary Fig. 1a, c)[19, 20]. The sequences of the N-terminal helices of E2s from yeast and human align well with highly conserved K or R residues at positions 5, 6, and 9 (UbcH7 numbering) (Supplementary Fig. 1f). Based on the sequence alignment, we mutated R5 and K9 in UbcH7 to Glu following the mutations in yeast xUbc1 and found the newly constructed xUbcH7 can pair with xUba1 to accept xUB transfer (Fig. 1a and Table 1). We have thus constructed an xUba1-xUbcH7 pair for xUB transfer through the OUT cascade. Since UbcH7 partners with HECT E3 in the cell, the exclusive delivery of xUB by xUba1

**Table 1 Mutants for the assembly of the OUT cascade with E6AP**

| | |
|---|---|
| xUB (human) | R42E, R72E |
| *xE1* | |
| xUba1 (yeast) | Q576R, S589R, D591R, E1004K, D1014K, E1016K |
| xUba1 (human) | Q608R, S621R, D623R, E1037K, D1047K, E1049K |
| *xE2* | |
| xUbc1 (yeast) | K5D, R6E, K9E, E10Q, Q12L |
| xUbcH7 (human) | R5E, K9E |
| *xE3* | |
| xE6AP (YW6) | D651R, D652E, M653W, M654H |

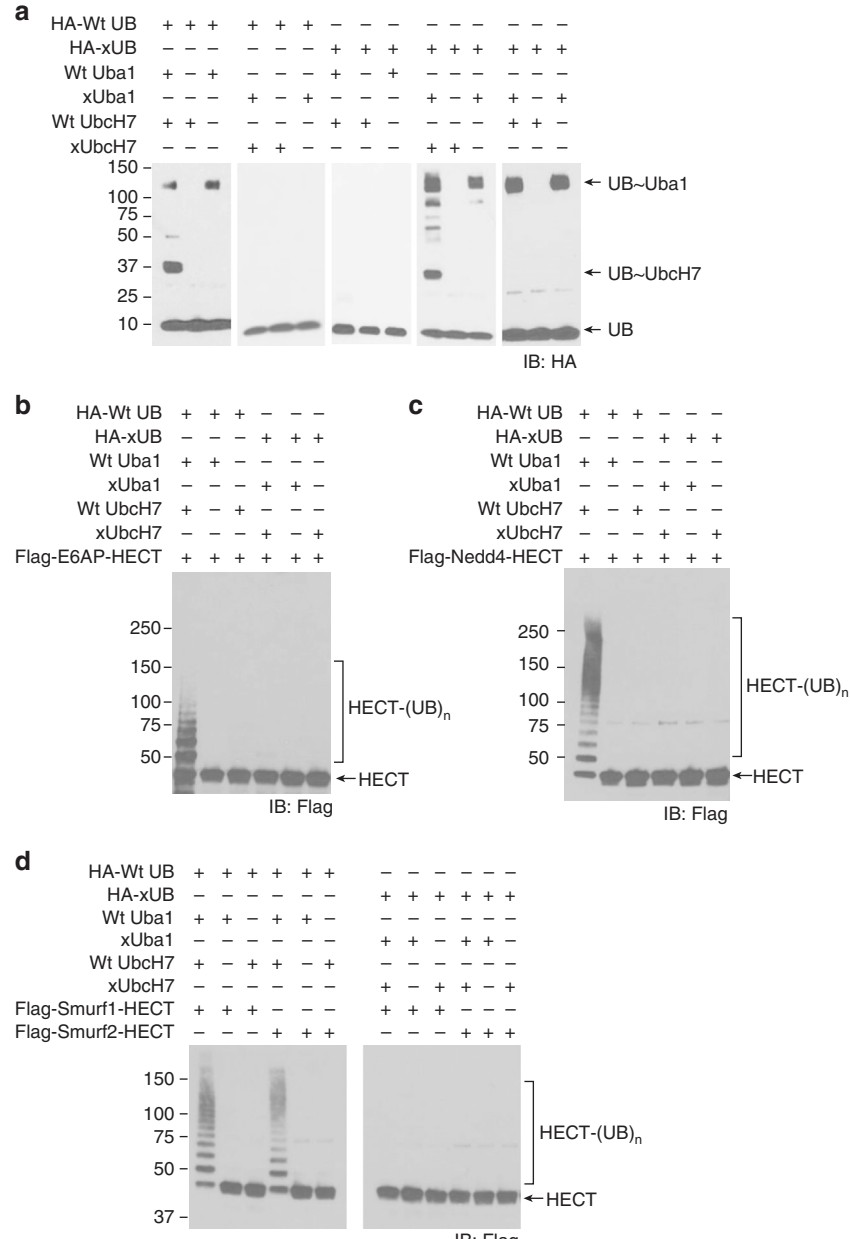

**Fig. 1** Orthogonality of the xUba1–xUbcH7 pair with the wt UB transferring enzymes. **a** Wt UB can be transferred to wt UbcH7 by wt human Uba1, but it cannot be activated by human xUba1 for transferring to xUbcH7. Vice versa, xUB can be activated by xUba1 for transferring to xUbcH7, but it cannot be activated by wt Uba1 for transferring to wt UbcH7, nor can it be transferred to wt UbcH7 by xUba1. The protein gels were run under non-reducing conditions to preserve the thioester conjugates of UB-E1 and UB-E2. **b–d** HECT domains of wt E6AP, Nedd4, Smurf1, and Smurf2 can be loaded with wt UB through the wt Uba1–UbcH7 pair. Yet they are not reactive with xUB though the xUba1–xUbcH7 pair. The protein gels were run under reducing conditions to probe the auto-ubiquitination of the HECT enzymes

to xUbcH7 paved the way for transferring xUB to a specific HECT E3 to profile its substrate proteins.

**Constructing the xUbcH7-xHECT pair with E6AP.** The N-terminal helix of UbcH7 is a key element not only for interaction with E1s but also for interaction with E3s (Fig. 2a, b). We found the R5E and K9E mutants in the N-terminal helix of xUbcH7 interfered with the transfer of xUB to wt HECT E3s such as E6AP, Nedd4, Smurf1, and Smurf2 (Fig. 1b–d). This is advantageous for the construction of the OUT cascade since it is preferred that xE2 would not pair with wt E3s to randomly transfer xUB to any E3 substrates in the cell. Our goal was to bridge xUB transfer through the last step of the OUT cascade by engineering

an orthogonal xUbcH7–xE6AP pair. For this purpose, we used yeast cell surface display to select for HECT mutants of E6AP that would restore binding with xUbcH7 to enable xUB loading on the HECT domain (Fig. 2c). For yeast selection, a HECT library of E6AP was expressed as fusions to the yeast cell surface protein Aga2P with each yeast cell displaying a specific member of the HECT library[21]. The yeast library was then reacted with biotin-labeled xUB, xUba1, and xUbcH7. HECT mutants catalytically active with xUbcH7 were loaded with xUB through the formation of xUB~HECT thioester conjugate. The catalytically active HECT mutants were further auto-ubiquitinated by xUB through Lys modification. As a result, the corresponding yeast cells were labeled with biotin that would bind to streptavidin conjugated

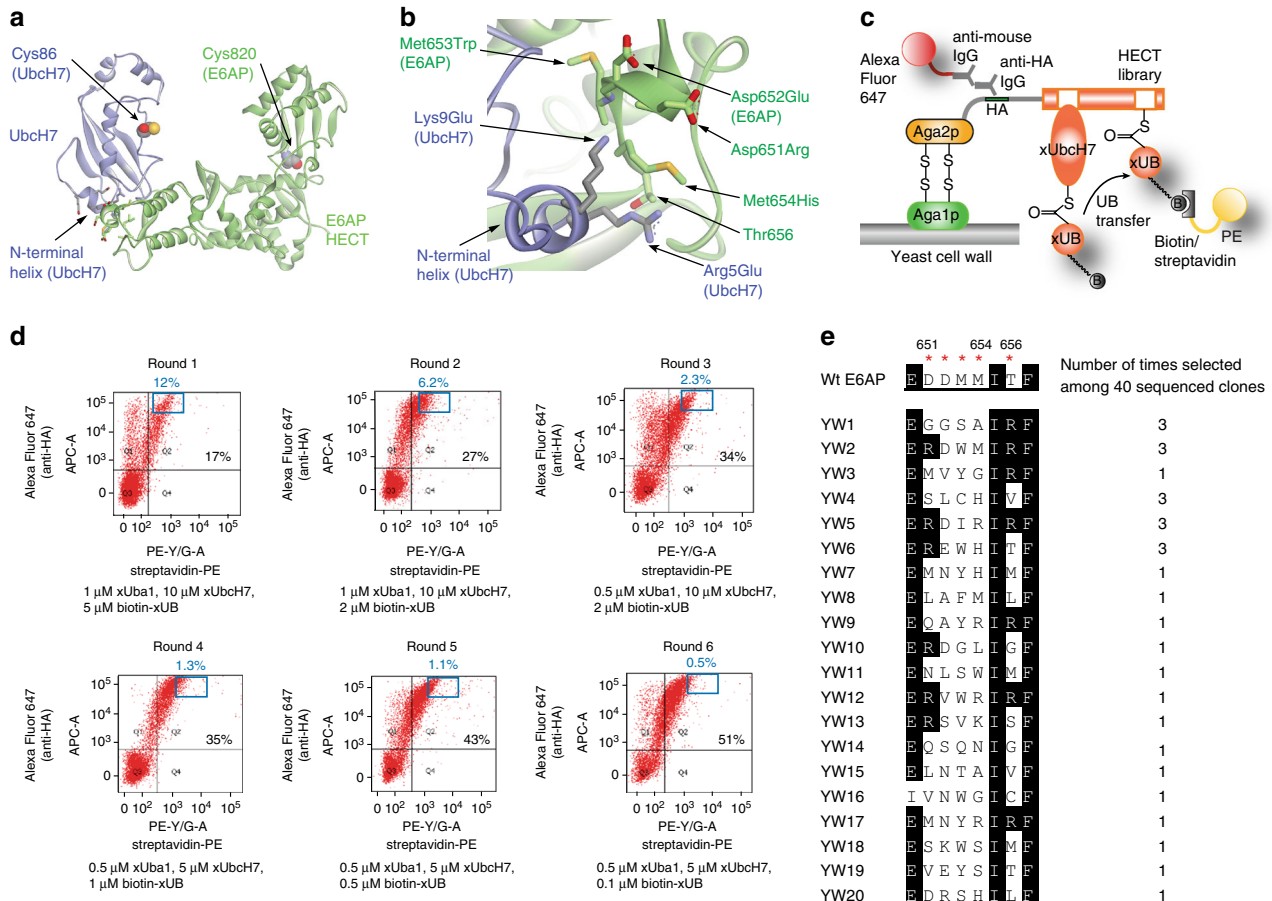

**Fig. 2** Yeast selection of E6AP library to engineer the xUbcH7-xE6AP pair. **a** Crystal structure of the E6AP HECT domain in complex with UbcH7 (PDB ID 1C4Z)[22]. The N-terminal helix of UbcH7 plays a key role in interacting with the HECT domain. The catalytic Cys residues in UbcH7 and the E6AP HECT are shown in CPK models. **b** Detailed interactions between the N-terminal helix of UbcH7 and the HECT domain of E6AP. To complement the R5E and K9E mutations in xUbcH7, D651, D652, M653, M654 and T656 in the HECT domain of E6AP were randomized for library selection by yeast cell surface display. HECT mutant YW6 with mutations D651R, D652E, M653W, and M654H was selected and used as xHECT in this study. **c** Yeast cell surface display of the HECT domain of E6AP and selection of the HECT library based on the transfer of biotin-xUB from xUbcH7 to the HECT domain. Biotin-xUB attached to HECT was labeled with streptavidin-PE, and the HA tag fused to the N-terminus of the HECT domain was labeled with a mouse anti-HA antibody and an anti-mouse IgG conjugated with Alexa 647. **d** FACS sorting of the HECT library of E6AP to select for yeast cells (EBY100) doubly labeled with PE and Alexa 647 as indicators of biotin-xUB conjugation and display of HECT on the cell surface, respectively. The HECT library underwent six rounds of xUB loading, streptavidin and antibody labeling, and cell sorting. Percentages in black designate fraction of yeast cells doubly labeled with PE and Alexa 647. The frames and percentages in blue designate the fraction of yeast cells collected by FACS in each round of selection. **e** Sequence alignment of the E6AP HECT clones selected by yeast cell surface display. Residues denoted by red stars were randomized in the HECT library

with phycoerythrin (PE). For selection of yeast cells displaying the HECT domain, a mouse anti-HA antibody was used to bind to the HA tag at the N terminus of the HECT and it was subsequently labeled with an anti-mouse IgG conjugated with Alexa 647. Fluorescence-activated cell sorting (FACS) was used to enrich the cells that were double labeled with the PE and Alexa 647. In this way, the sorting was to enrich cells that displayed catalytically active HECT domains capable of bridging xUB transfer from xUbcH7. We validated the selection protocol by displaying the wt HECT domain of E6AP on the yeast cell surface, and demonstrating the efficient labeling of the yeast cells by biotin-wt UB transferred through the wt Uba1−UbcH7 pair (Supplementary Fig. 2).

We constructed a HECT library of E6AP based on the crystal structure of the HECT domain with UbcH7 (Fig. 2a, b)[22]. Residues R5 and K9 in the N-terminal helix of UbcH7 were mutated to Glu in xUbcH7 and the crystal structure shows that these residues mainly interact with a helical turn in the HECT domain of E6AP. R5 of UbcH7 is in close distance (4.6 Å) with

the hydroxyl group of T656 of HECT. It also packs on the side chain of M654 that is 3.2 Å away. K9 of UbcH7 may form salt bridges with HECT D651 and D652 that are a short distance apart (4.5 Å). We thus decided to randomize E6AP residues D651, D652, M653, M654, and T656 to generate the HECT library.

We expressed the library on the yeast cell surface and carried out the selection by transferring biotin-xUB to the HECT mutants through the xUba1-xUbcH7 pair. Cells were labeled with streptavidin and antibody conjugates with fluorophores as in the model selection, and FACS was performed to harvest cells that were double labeled with PE and Alexa 647. Cells collected were cultured for next round of biotin-xUB loading, fluorescent labeling, and FACS. After 6 rounds of sorting, 51% of the cells were double labeled with both fluorophores suggesting a population of HECT mutants with efficient xUB transfer activity from xUbcH7 were selected (Fig. 2d). DNA sequencing of the 40 clones from the 6th round of sorting showed a clear pattern of convergence (Fig. 2e). Clones appearing multiple times tend to

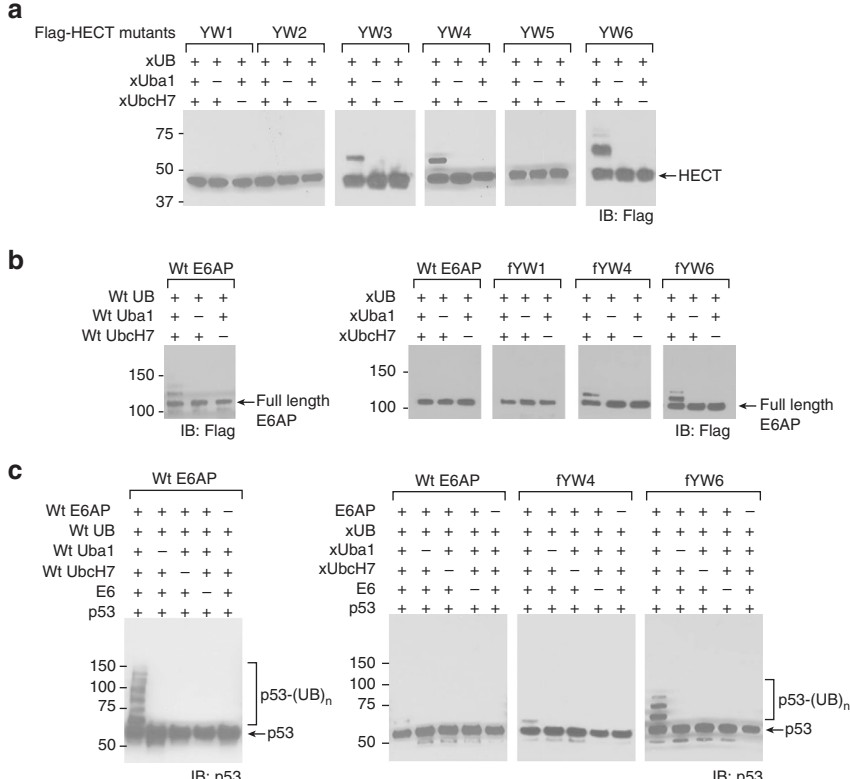

**Fig. 3** Activity in xUB transfer by the E6AP mutants identified from yeast cell selection. **a** Activity of the HECT mutants YW1–6 in auto-ubiquitination by xUB through the xUba1–xUbcH7 pair. **b** Activity of the full-length E6AP mutants fYW1, fYW4, and fYW6 in auto-ubiquitination by the xUba1–xUbcH7 pair (right panel). While wt E6AP can be auto-ubiquitinated by the wt Uba1–UbcH7 pair (left panel), it cannot be conjugated to xUB by the xUba1–xUbcH7 pair (right panel). **c** p53 ubiquitination by wt E6AP and fYW4 and fYW6 in the presence of E6. p53 can be efficiently ubiquitinated by wt UB through the wt Uba1–UbcH7–E6AP cascade in the presence of E6 (left panel). p53 can also be efficiently modified by xUB through the engineered cascade of xUba1–xUbcH7–fYW6 in the presence of E6 (right panels). In contrast, xUB has very low activity in ubiquitinating p53 through the crossover cascade of xUba1–xUbcH7–wt E6AP in the presence of E6. All blots are representative of at least three independent experiments

have D651 in HECT replaced with an Arg (YW2, YW5, and YW6). This change matches the charge reverse mutation of K9E in xUbcH7 (Fig. 2b). D652 of HECT, although randomized in the library, was most often unchanged or replaced with a similar Glu in the selected clones (YW2, YW5, and YW6). M653 is often replaced by aromatic residues such as Tyr and Trp in the selected clones (YW2, YW3, and YW6). M654 is replaced by positively charged Arg or His residues (YW4, YW5 and YW6), and residues selected at T656 is also quite converged showing a preference for positively charged Arg (YW1–3 and YW5). These changes match well with the charge reversal of R5E mutation in xUbcH7. We thus assayed if the individual HECT mutants from FACS could mediate xUB transfer with the xUba1–xUbcH7 pair.

**Verifying xUB transfer through E6AP mutants.** We separately cultured yeast clones YW1-6 and reacted the yeast cells with the xUba1-xUbcH7 pair for biotin-xUB loading (Supplementary Fig. 3). We found yeast clones YW4 and YW6 had the strongest loading of biotin-xUB with 25% and 34% of the cells doubly labeled. To check the activities of individual HECT domains, we expressed mutants YW1-6 in *E. coli* and found YW3, YW4, and YW6 could be efficiently auto-ubiquitinated with xUB through the xUba1–xUbcH7 pair while YW1, YW2, and YW5 were not active for xUB transfer (Fig. 3a). We suspect the difference in activities of the HECT domains anchored on yeast cell surface and free in solution may be due to the change in their folding status in different environments. We replaced the wt HECT domain in E6AP with the mutant HECT of YW1, YW4, and YW6

to generate the full-length E6AP mutants fYW1, fYW4, and fYW6. We found fYW4 and fYW6 can be auto-ubiquitinated by xUB through the xUba1–xUbcH7 cascade (Fig. 3b). However, wt E6AP and fYW1 were not active in auto-ubiquitination by xUB in combination with the xUba1–xUbcH7 pair. We then tested the transfer of xUB from the E6AP mutants to p53, a key ubiquitination target recruited by E6[11, 16]. We found fYW6 could efficiently ubiquitinate p53 with xUB through the xUba1–xUbcH7–fYW6 cascade and the ubiquitination was dependent on the E6 protein (Fig. 3c). The activity of xUB transfer to p53 through fYW6 was approaching the activity of wt UB transfer through wt E6AP. In contrast, when the xUba1–xUbcH7 pair was reacted with wt E6AP, we only observed very low activity in transferring xUB to p53 suggesting the minimal cross-reactivity of xUB with native UB transfer pathways (Fig. 3c). Comparing to fYW6, fYW4 was less active in transferring xUB to p53. We thus decided to use fYW6 as xE6AP in the OUT cascade to identify E6AP substrates in the cell (Table 1).

**Verifying the orthogonality of OUT cascade in cells.** We next tested if xUB could be exclusively transferred through the OUT cascade of xUba1-xUbcH7-xE6AP in the cells without crossing over to the wt UB transfer cascades (Fig. 4a). We constructed a lentiviral vector to express wt UB or xUB with tandem 6× His and biotin tags at the N terminus of UB (HBT-wt UB and HBT-xUB)[23]. We also screened HEK293 cells that stably expressed wt Uba1 or xUba1 with an N-terminal Flag tag. Transient expression of HBT-wt UB and HBT-xUB in these cells followed by affinity pull-down

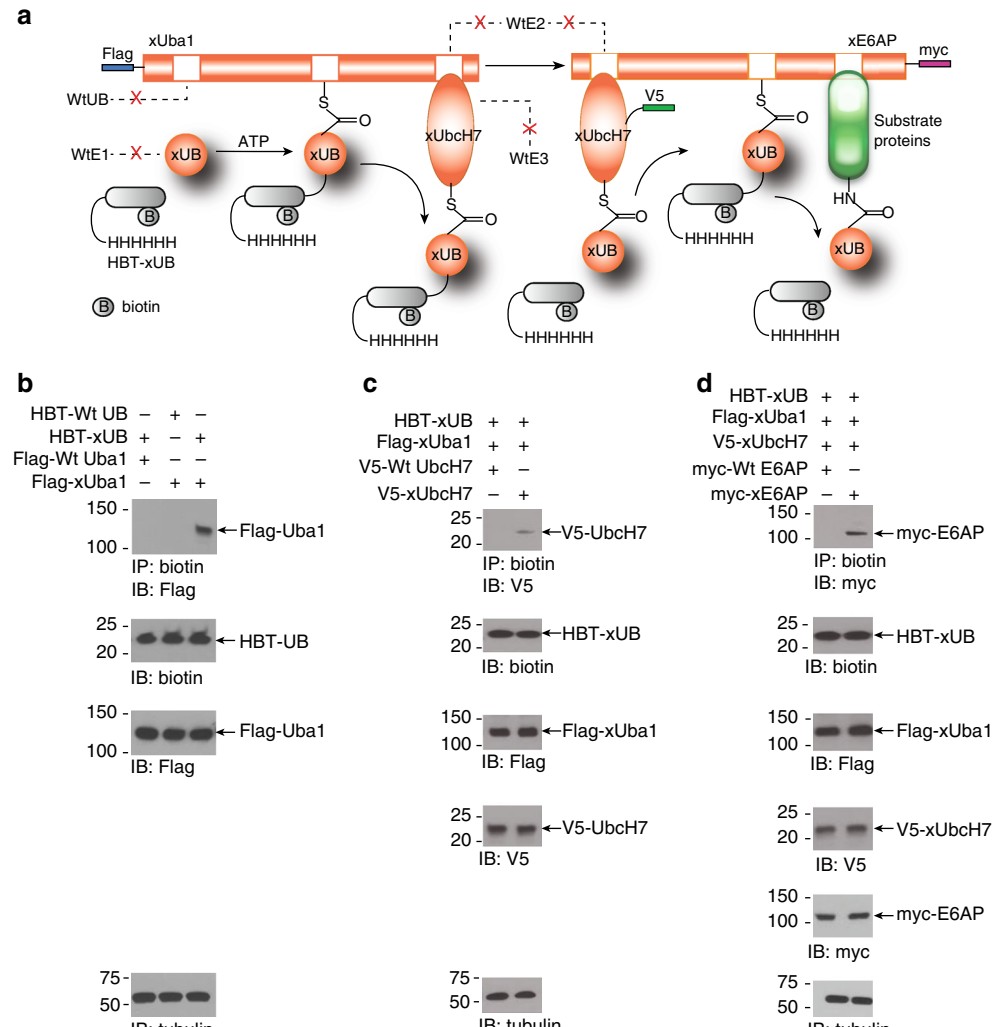

**Fig. 4** Orthogonality of the E6AP OUT cascade with wt Uba1-UbcH7-E6AP cascade in HEK293 cells. **a** Scheme showing exclusive transfer of xUB through the OUT cascade of E6AP. xUba1, xUbcH7 and xE6AP are tagged with Flag, V5 and myc tags, respectively. Dotted lines with a red "X" designate none interactions between OUT components and the native E1, E2, and E3 enzymes. xUB is tagged with biotin and 6× His tags (HBT-xUB). **b** xUB and xUba1 formed xUB–xUba1 conjugate in HEK293 cells. As a result of conjugate formation, flag-xUba1 could be copurified with HBT-xUB by binding to streptavidin beads. In contrast, flag-wt Uba1 could not be copurified with HBT-xUB, neither could flag-xUba1 be copurified with HBT-wt UB in the cells co-expressing the UB-E1 pairs. This suggests that the crossover xUB-wt Uba1 or wt UB-xUba1 pairs are not active in the cell. **c** xUB could be transferred through xUba1-xUbcH7 pair to form xUB–xUbcH7 conjugate in the cell. Due to the conjugate formation, V5-xUbcH7 could be copurified with HBT-xUB from cells expressing HBT-xUB, flag-xUba1 and V5-xUbcH7. In contrast, the expression of HBT-xUB and the crossover pair of xUba1-wt UbcH7 would not generate xUB–wt UbcH7 conjugate, so V5-wt UbcH7 could not be co-purified with HBT-xUB. **d** xUB could be transferred through xUba1–xUbcH7–xE6AP cascade to form xUB–xE6AP conjugate in the cell. As a result, myc-xE6AP could be co-purified with HBT-xUB in cells expressing HBT-xUB and the xUba1–xUbcH7–xE6AP cascade. In contrast, coexpression of HBT-xUB with the crossover cascade of xUba1–xUbcH7–wt E6AP would not generate xUB–wt E6AP conjugate, so myc-wt E6AP could not be co-purified with HBT-xUB

with streptavidin beads showed that xUba1 was co-precipitated with HBT-xUB, but wt Uba1 could not be co-precipitated with HBT-xUB, neither xUba1 could be co-precipitated with HBT-wt UB (Fig. 4b). This suggests that xUB was exclusively reactive with xUba1 in the cell, and there is no cross activities between xUB and wt Uba1, or between wt UB and xUba1. To probe the orthogonality at the E1–E2 interface, we co-expressed HBT-xUB with either wt UbcH7 or xUbcH7 in cells stably expressing xUba1. We found V5-tagged xUbcH7 could be purified with the streptavidin beads suggesting the formation of HBT-xUB–xUbcH7 conjugate, yet V5-tagged wt UbcH7 could not be co-purified with HBT-xUB (Fig. 4c). This proves that xUba1-xUbcH7 was a functional relay for xUB in the cell, but xUba1–wt UbcH7 pair could not mediate xUB transfer to a wt E2. To verify the orthogonality at the E2-E3 interface, we co-expressed HBT-

xUB with either wt E6AP or xE6AP in HEK293 cells stably expressing the xUba1-xUbcH7 pair, and purified the xUB-conjugated proteins by streptavidin beads. Myc-tagged xE6AP was co-purified with xUB suggesting the formation of xUB–xE6AP conjugate, yet no wt E6AP was conjugated with xUB (Fig. 4d). These results prove that xUba1–xUbcH7–xE6AP is an orthogonal cascade for the transfer of xUB, and the crossover of xUB to wt cascades was eliminated.

**Profiling the substrates of E6AP in HEK293 cells by OUT.** To express the OUT cascade of E6AP in the cell, we screened cell lines stably expressing xUba1, xUbcH7 and xE6AP by lentiviral infection. Western blots of the cell lysate probed with antibodies against each OUT component suggested their adequate

expression (Supplementary Fig. 4b). We also generated a stable cell line expressing the xUba1–xUbcH7 cascade without xE6AP as a control for the proteomic screen. To initiate xUB transfer through the OUT cascade, we transduced the two stable cell lines with lentivirus carrying the vector to express HBT-xUB. We then purified cellular proteins conjugated with HBT-xUB sequentially by Ni-NTA and streptavidin affinity columns under strong denaturing conditions (Supplementary Fig. 4a). We found xUba1, xUbc1, and xE6AP are among the proteins retained by tandem purification suggesting the loading of HBT-xUB to the engineered E1, E2, and E3 enzymes of the OUT cascade (Supplementary Fig. 4c). We then digested the proteins on the streptavidin beads by trypsin and analyzed the peptide fragments by LC-MS/MS to identify xUB-conjugated proteins. In parallel, we performed tandem purification and proteomic analysis on control cells expressing the xUba1–xUbcH7 pair without xE6AP (Supplementary Fig. 4d, e). By comparing the two proteomic profiles, we identified proteins that had ratios of peptide-spectrum match (PSM) 2-fold or higher between cells expressing the full E6AP OUT cascade and the control cells. We carried out affinity purification and proteomic screen three times. We found 130 proteins repeatedly appearing in all three screens with PSM ratio ≥ 2 (Supplementary Data 1). These proteins are likely the direct ubiquitination targets of E6AP.

Among the E6AP targets identified, we found previously identified substrates such as the UV excision repair protein HHR23A (RAD23A) and HHR23B (RAD23B), proteasomal ubiquitin receptor ADRM1, 26S proteasome non-ATPase regulatory subunit 4 (PSMD4 or Rpn10), 26S proteasome AAA-ATPase subunit Rpt5 (PSMC3), and E3 ligase RING2 (RNF2 or Ring1B)[24–27]. Ingenuity Pathway Analysis (IPA) of the proteins from the OUT screen showed that E6AP targets have a significant association with a variety of canonical pathways (Supplementary Data 2). It is intriguing that several associated pathways mediate cell cycle control and chromosome replication, matching the role of E6AP in viral oncogenesis. IPA also identified 8 protein networks that are significantly associated with E6AP substrates (Supplementary Data 3). The identified networks are related to cell death and survival, DNA replication, recombination, and repair, cellular growth and proliferation, and nervous system development and function.

We used the CRAPome database to evaluate whether proteins non-specifically bound to the affinity resins were among the targets identified by OUT. CRAPome selects non-specific binders in proteomic experiments based on the frequency of their appearance in pull-down experiments with various bait proteins under non-denaturing conditions[28]. In contrast, we used strong denaturing condition to purify xUB-conjugated proteins in the OUT screen. Nevertheless, we found 2 of the 130 E6AP targets identified by OUT have a frequency higher than 34% in the CRAPome database (Supplementary Data 4). We verified that one of them, PRMT5, is a E6AP target (see below). We also repeatedly identified 35 proteins in control cells without expression of xE6AP, and they were not present among xUB-conjugated proteins purified from cells expressing the full OUT cascade of E6AP (Supplementary Data 5).

**In vitro and in vivo validation of the E6AP substrates**. We found some key signaling enzymes such as kinases MAPK1, CDK1, CDK4, protein Arg methyltransferase 5 (PRMT5), transcription factor β-catenin, and FAS-associated factor UbxD8 are likely substrates of E6AP (Supplementary Data 1). We thus assayed if E6AP targets them for ubiquitination and regulates their stabilities in the cell. We first used the wt UB transfer cascade Uba1-UbcH7-E6AP to test if the potential substrate proteins

could be modified by wt UB in vitro. We expressed and purified the potential substrates from *E. coli* cells, and found that they were ubiquitinated by E6AP to different extents: CDK1, CDK4 and β-catenin were strongly ubiquitinated by E6AP with the formation of high molecular weight bands, while MAPK1, PRMT5, and UbxD8 mainly generated species with one or two conjugated UBs (Fig. 5). As a positive control, E6AP-catalyzed ubiquitination of HHR23A, a previously identified E6AP substrate, was confirmed (Fig. 5g)[26]. Protein expressed in E. coli cells may not bear the proper posttranslational modifications for E6AP recognition, so the in vitro assays may not reflect the real activity of E6AP with the substrate proteins. We thus tested whether the potential substrates are targeted for ubiquitination by E6AP in the cell.

We inhibited E6AP expression in HEK293 cells with lentivirus delivering the anti-E6AP shRNA. We also overexpressed E6AP in blank HEK293 cells and cells harboring the anti-E6AP shRNA. Cells were treated with proteasome inhibitor MG132 before harvesting to inhibit protein degradation. Ubiquitination levels of various substrates in different cell populations were revealed by immunoprecipitation with substrate-specific antibodies and immunoblotting with an anti-UB antibody. Comparing to the parental HEK293 cells, cells expressing anti-E6AP shRNA had significantly lower levels of poly-ubiquitinated forms of MAPK1, CDK1, CDK4, PRMT5, β-catenin, and UbxD8 (Fig. 6). The poly-ubiquitination of each target protein in the HEK293 cells harboring the anti-E6AP shRNA can be restored by over-expressing E6AP in the cell. Furthermore, HEK293 cells with over-expression of E6AP gave rise to more intense poly-ubiquitination of MAPK1, CDK1, CDK4, β-catenin, and UbxD8 comparing to the parental HEK293 cells. The known E6AP substrate HHR23A showed similar dependence on E6AP for its ubiquitination in the HEK293 cell. These results prove that the potential E6AP substrates identified by the OUT screen are indeed E6AP targets in the cell.

To probe whether E6AP-mediated ubiquitination would signal protein degradation, we transiently transfected HEK293 cells with varying amounts of wt E6AP plasmid. Western blot of the cell lysates with an anti-E6AP antibody showed an increased E6AP expression in the cells receiving more plasmid DNA. In parallel, the levels of CDK1, CDK4, PRMT5, and UbxD8 were significantly reduced in comparison to the control cells without transfection of the E6AP plasmid (Fig. 7a, b). We also measured the half-lives of the target proteins with the cycloheximide (CHX) chase assay. Cells expressing E6AP were treated with CHX to inhibit protein synthesis and the substrate levels in the cell were measured by immunoblotting with anti-substrate antibodies at different time points. We found that the turnover of PRMT5, CDK1, CDK4, and UbxD8 was significantly faster in cells overexpressing E6AP compared to control cells transfected with the same amount of empty plasmid (Fig. 7c, d). The level of MAPK1 remained stable despite over-expression of E6AP. MAPK1 is among the 400 most expressed proteins in the cell and its half-life is longer than 68 h[29, 30]. This may explain why E6AP expression had little effect on the level of MAPK1 in the cell. On the other hand, inhibiting endogenous E6AP expression in HEK293 cells by shRNA stabilized PRMT5, CDK1, CDK4, β-catenin, UbxD8 (Fig. 7c, d). The known E6AP substrate HHR23A was also stabilized with the decreased expression of E6AP.

## Discussion

The large number of E3s (>600) encoded in the human genome reflects the key roles they play in cell regulation. On the other hand, their diversity makes it a significant challenge to identify the direct substrates of individual E3s. Current methods screening

E3 substrates fall into three categories—affinity binding to E3, monitoring changes in protein stability or ubiquitination levels in response to E3 perturbation, or trapping E3 substrates by covalent or noncovalent interactions (Supplementary Fig. 5). Affinity-based approaches such as co-immunoprecipitation, yeast two-hybrid system, and protein microarray have been used to screen E3 substrates based on the binding between E3 and substrates (Supplementary Fig. 5a)[31–33]. They are less specific since the $K_d$'s of the E3-substrate complexes are around hundreds of μM, and the complexes are transient[34]. Also, proteins other than substrates can bind to E3s to function as adaptors or regulators. Still, these methods yielded important substrate profiles of HECT E3 E6AP, Ring E3 anaphase-promoting complex (APC), and the Skp1-cullin-F-box (SCF) complex[31–33]. A more direct approach to assign E3 substrates is to correlate changes in E3 activity with the changes in stability or ubiquitination level of cellular proteins (Supplementary Fig. 5b). One method known as "global protein stability profiling (GPS)" tracks the stability of thousands of proteins with a fused fluorescence protein tag, and it has been used to screen substrates of SCF E3s[35–37]. The development of anti-diGly antibody allows affinity enrichment of substrate peptides containing the ubiquitination sites, and comparison of protein ubiquitination levels upon perturbation of E3 activity[38]. Using the quantitative diGly proteomics (QdiGly), substrate profiles of cullin-Ring and Parkin E3s were generated[39, 40]. E3s are also converted to substrate traps so the substrate proteins would still be bound to E3 after UB transfer. This would enable the co-purification of the substrate proteins with E3s. To create "UB-activated interaction traps (UBAIT)", UB was fused to HECT and Ring E3s and it can attack the substrates bound to the E3-UB fusion to generate covalent linkages between E3 and the substrate proteins (Supplementary Fig. 5c)[41]. Another design is to fuse the F-box proteins with a series of UB associated domain (UBA) and use them as UB ligase traps[42, 43]. As F-box proteins recruit substrates to SCF E3s, the UB chain extending from the substrate would bind to the UBA repeats with high affinity. Purification of proteins bound to F-box-UBA fusion would enrich the substrates of the F-box protein (Supplementary Fig. 5d). E2-E3 fusions has also been used to identify E3 substrates. Ubc12, the E2 enzyme mediating Nedd8 transfer, was fused to the substrate binding domain of Ring E3 XIAP. The fusion protein, known as a "Neddylator", allows Nedd8 transfer to XIAP substrates so the ubiquitination targets of XIAP could be identified among Nedd8-modified proteins (Supplementary Fig. 5e)[44]. The development of diverse methods to profile E3 substrates enables the interrogation of E3 function from different perspectives. The substrate profiles generated by various methods could corroborate to reveal the functions of E3s.

Here, we developed a method known as "orthogonal ubiquitin transfer (OUT)" to identify the direct substrates of HECT E3 E6AP (Fig. 4a). In OUT, an affinity-tagged UB variant (xUB) is exclusively transferred through an engineered xE1-xE2-xE3 cascade to the substrates of a specific E3. By purifying xUB-modified proteins from the cell and identifying them by proteomics, we would be able to identify the direct substrates of a E3. In this study, we used OUT to identify 130 potential E6AP substrates, and among them, we confirmed MAPK1, PMRT5, CDK1, CDK4, β-catenin, and UbxD8 are ubiquitinated by E6AP in the HEK293 cells. During the revision of this manuscript, β-catenin was confirmed as an E6AP substrate by another report[45]. A key advantage of OUT is that it assigns E3 substrates by directly following UB transfer from the E3 to its substrate proteins. Methods based on E3 substrate binding, or change of protein stability upon perturbation of E3 activity, use indirect readouts of substrate ubiquitination to assign E3 substrates. The substrate profiles generated by those methods could be distorted by factors

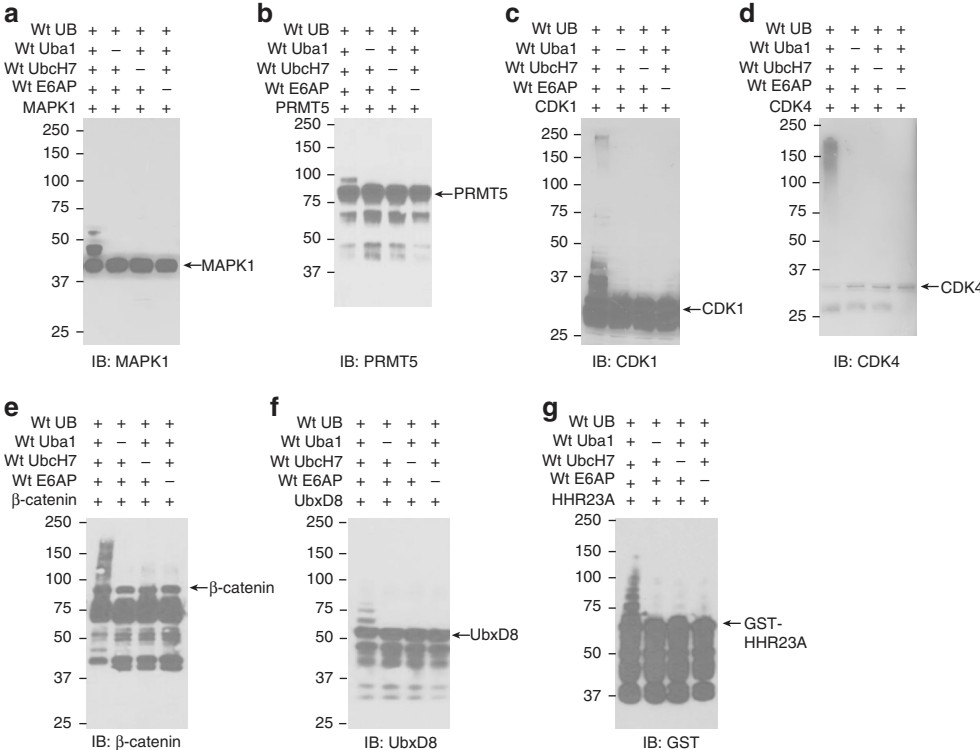

**Fig. 5** In vitro assays to test ubiquitination of E6AP substrates identified by OUT. wt UB was transferred through wt Uba1-UbcH7-E6AP cascade to the potential substrate proteins expressed from *E. coli* cells. E6AP ubiquitination of MAPK1 **a**, PRMT5 **b**, CDK1 **c**, CDK4 **d**, β-catenin **e**, and UbxD8 **f**, were confirmed. E6AP-catalyzed ubiquitination of HHR23A, a previously reported E6AP substrate, was also assayed **g**. All blots are representative of at least three independent experiments

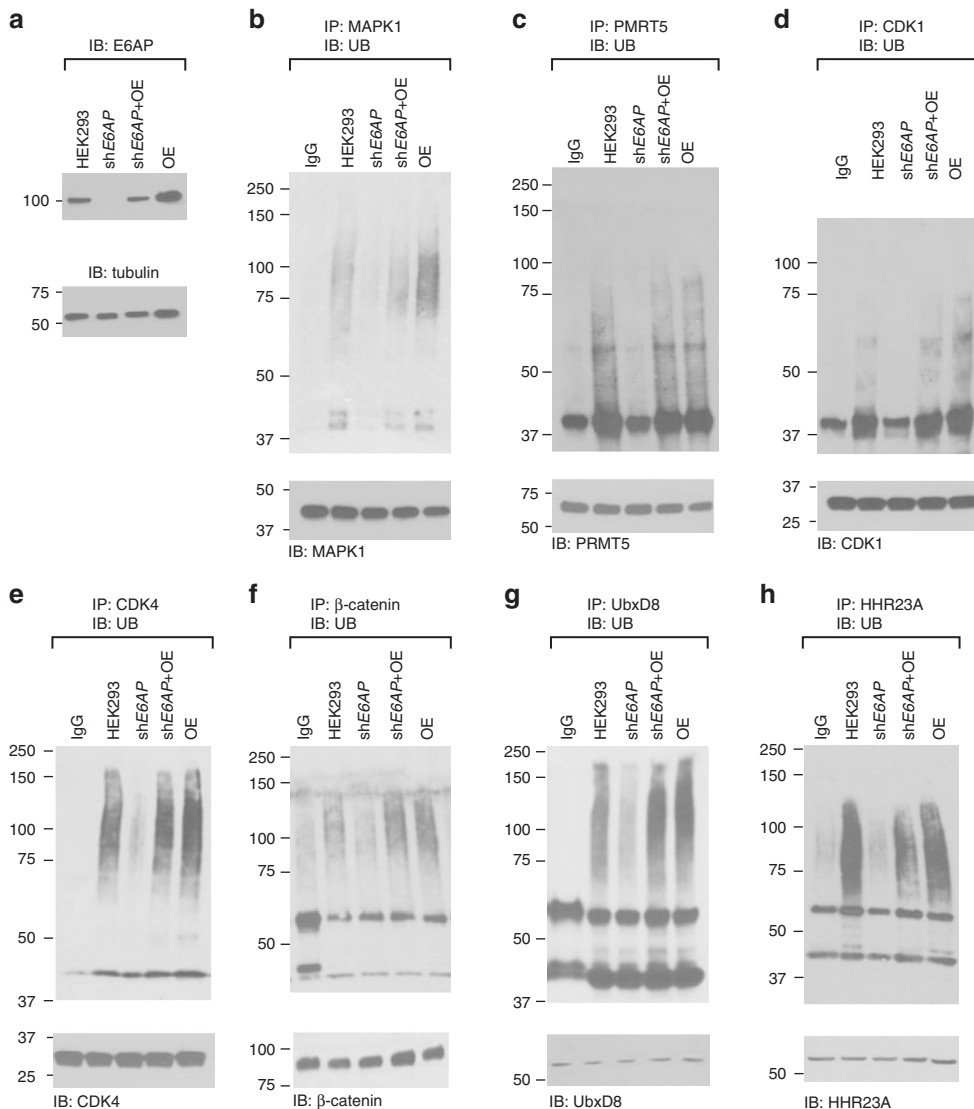

**Fig. 6** Cellular assays to test the ubiquitination of E6AP substrates identified by OUT. **a** Inhibition of E6AP expression in HEK293 cells by sh*E6AP* was confirmed with Western blot probed with an antibody against E6AP. **b–g** Ubiquitination of MAPK1 **b**, PRMT5 **c**, CDK1 **d**,CDK4 **e**, β-catenin **f** and UbxD8 **g** in HECK293 cells was assayed by immunoprecipitation with antibodies against each substrate proteins and probing the ubiquitination levels of the proteins with an anti-UB antibody on the western blots. After 1.5-h treatment of cells with MG132, ubiquitination of each target protein was compared among the blank HEK293 cell (HEK293), HEK293 expressing sh*E6AP* (sh*E6AP*), HEK293 expressing both sh*E6AP* and recombinant E6AP (sh*E6AP* + OE), and HEK293 expressing recombinant E6AP (OE). **h** Ubiquitination of HHR23A, a known E6AP substrate, was assayed as a control. Rabbit IgG was used as a control for immunoprecipitation in **b** and **e**. Mouse IgG was used as a control for immunoprecipitation in **c**,**d** and **f**–**h**. All blots are representative of at least three independent experiments

such as the binding of adaptor or regulatory proteins to E3, or the attachment of UB chains of non-degradation signals to the substrates. E3 may also regulate the activities of proteasome and other E3s, thus perturbing the activity of one E3 may affect the degradation or ubiquitination levels of the substrates of other E3s[24, 46–48]. In this study, we used yeast cell surface display to identify mutations at the E2-binding site of the E6AP HECT domain to generate an xE2–xE3 pair for the OUT cascade. The mutations at the E2-binding site of E6AP shall have minimal disturbance to its substrate profile. In comparison with various E3 fusions as substrate traps, the engineered xE6AP would be a better reenactor of the wt E3 in transferring xUB to the substrate proteins to enable their identification by OUT.

OUT has a few limitations. First, each E3 would require its own OUT cascade for substrate identification. The xUB-xE1 pair we engineered can be used for the OUT cascade of various E3s. Due to the high sequence homology of the N-terminal helices of the E2s, mutations can be transplanted from xUbc1 to UbcH7 and UbcH5b to generate xE1–xE2 pairs[17]. The great diversity of E3s would require the engineering of individual E3s to assemble xE2–xE3 pairs. Here we used yeast cell surface display to identify HECT mutants of E6AP that can pair with xUbcH7. The helical turn we randomized in the E6AP HECT domain is a common element in many HECT enzymes[49–52]. It is possible to generate orthogonal xE2-xE3 pairs by transplanting the mutations from xE6AP to other HECTs E3s. If such strategy is not effective, the yeast selection platform for E6AP HECT could be used to engineer other HECTs such as Smurf1/2, Nedd4-1/2, and Huwe1, all playing important roles in cell regulation[5]. Another limitation of OUT is that co-expression of HBT-xUB and the full xE1–xE2–xE3 cascade, although successful in HEK293 cells, maybe a challenge in other cell types. The recently developed

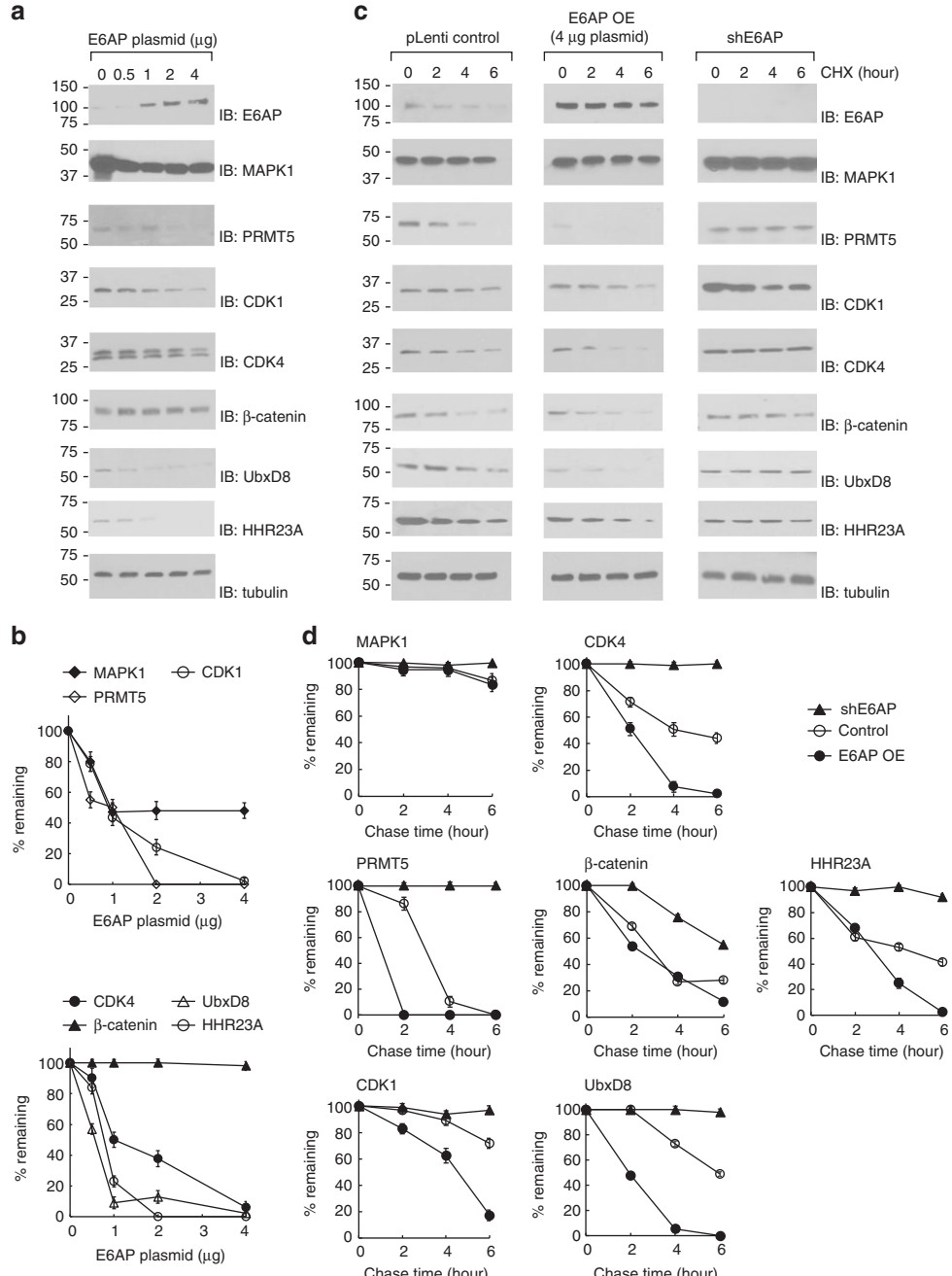

**Fig. 7** Effect of E6AP expression on the stability of the substrate proteins. **a** E6AP decreases the steady-state levels of the substrate proteins in the HEK293 cells. Cells were transfected with increasing amount of E6AP plasmid. Levels of the E6AP substrates were assayed with immunoblots of the cell lysate probed with substrate-specific antibodies. Approximately $5 \times 10^6$ cells were used for each transfection of the E6AP plasmid. **b** Quantitative analysis of substrate levels in correlation with E6AP expression. Intensity of the bands in **a** were plotted against the amount of pLenti E6AP plasmid used for transfection assuming 100% of substrate protein when an empty plasmid was used for mock transfection. Results were the average of three repeats. **c** E6AP-dependent degradation of substrate proteins assayed by cycloheximide (CHX) chase. HEK293 cells ($5 \times 10^6$ cells) were transfected with 4 μg of pLenti E6AP plasmid with the same amount of empty pLenti plasmid used in the controls. The cells were treated with 100 mg/ml CHX 48 h after transfection. Cell extracts were collected at 0, 2, 4, and 6 h after incubation with CHX, followed by immunoblotting with substrate-specific antibodies. **d** Quantitative analysis of the levels of the substrate proteins in the cell in the CHX chase experiment. Data are representative of three independent experiments. Besides the new substrates identified by OUT, HHR23A, a known E6AP substrate, was assayed for its degradation regulated by E6AP. The vertical bars in **b** and **d** represent s.e.m. from three independent experiments

genome editing tools such as CRISPR/Cas9 may provide an opportunity to introduce the OUT cascade into the original genetic background to identify E3 substrates[53].

We found E6AP expression did not affect the stability of MAPK1. The ubiquitination and degradation UbxD8 signaled by

E6AP suggests a role for E6AP in lipid metabolism, since UbxD8, by forming a complex with p97/VCP, regulates lipid droplet size and abundance[54]. Consistently, expression of a dominant-negative E6AP mutant promotes accumulation of lipid droplets[55], which may involve stabilized UbxD8 due to decreased

activity of E6AP. The biological significance of E6AP-mediated ubiquitination of CDK1, CDK4 and PRMT5 awaits further investigations. Recent studies have suggested that E6AP is required for cellular senescence, i.e., irreversible exit from the cell cycle, as a physiological response to oxidative stress or oncogene activation[56–58]. Thus, E6AP-mediated ubiquitination of CDKs in the absence of viral oncoproteins may be involved in the senescence response to various cellular stresses. Indeed, Ingenuity Pathway Analysis showed that many of the potential E6AP substrates identified by OUT are associated with pathways and networks relevant to DNA replication, cell cycle control, oncogenic signaling, cell survival/death and development (Supplementary Data 2 and 3). PRMT5 plays a key role in chromatin regulation by methylating Arg residues in histones[59]. Since various studies have indicated the significance of epigenetic changes in cancers and autism spectrum diseases, it would be interesting to determine how E6AP-mediated PRMT5 ubiquitination is involved in the pathobiology of those diseases[60–62].

## Methods

**Reagents**. XL1 Blue cells were from Agilent Technologies (Santa Clara, CA, USA). pET-15b and pET-28a plasmids for protein expression were from Novagen (Madison, WI, USA). pCTCON2 plasmid and the yeast strain EBY100 were from K. Dane Wittrup of Massachusetts Institute of Technology[21]. The plasmid with the human Uba1 gene was from Wade Harper of Harvard Medical School[8]. The plasmid for E6AP expression was from Jon M. Huibregtse of the University of Texas at Austin[63]. The plasmid for Rsp5 expression was from Linda Hicke of the Northwestern University[64]. pQCXIP HBT-Ubiquitin (26865) was from Addgene (Cambridge, MA, USA)[23]. pLenti-puro plasmid was from Addgene (39478). pLenti4/V5-DEST-zeocin (K498000) and ViraPower Lentiviral Packaging Mix (K4975-00) were from Life Technology. pLenti-puro plasmid was from Addgene (39478). pET-PRMT5 plasmid was provided by Yujun Zheng of University of Georgia, Athens. The mammalian cell expression vectors for MAPK1 (39230), CDK1 (27652), and UbxD8 (53777) and pGEX-HHR23A (10864) were from Addgene. HEK293 cells were from American Tissue Culture Collection (ATCC), and cultured in high-glucose Dulbecco's modified Eagles medium (DMEM) (Life Technologies, Carlsbad, CA, USA, 11965092) with 10% (v/v) Fetal bovine serum (FBS) (Life Technologies, 11965092). Antibiotics hygromycin, blasticidin, zeocin and puromycin were from GiBCO/Invitrogen (Carlsbad, CA, USA) and RPI (Mount Prospect, IL, USA). Doxycycline was from RPI.

Anti-β-catenin antibody (sc-65480), anti-CDK1 antibody (sc-54), anti-CDK4 antibody (sc-260), anti-E6AP antibody (sc-25509), anti-HA antibody (sc-7392), anti-HHR23A antibody (sc-365669), anti-MAPK1 antibody (sc-154), anti-Myc antibody (sc-40), anti-p53 antibody (sc-126), anti-PRMT5 antibody (sc-376937), anti-α-Tubulin antibody (sc-23948), anti-V5 antibody (sc-271944), anti-UB antibody (sc-8017), anti-UbxD8 antibody (sc-374098) were from Santa Cruz Biotechnology. These antibodies were diluted between 500 and 1000-fold to probe the Western blots. Goat anti-rabbit IgG-HRP (sc-2004) and goat anti-rabbit IgG-HRP (sc-2005) were also from Santa Cruz Biotechnology and were diluted 10,000-fold as the secondary antibody for western blotting. Streptavidin-HRP-conjugate was from Life Technologies and was diluted 20,000-fold for western blotting. Anti-Flag M2 antibody (F3165) was from Sigma-Aldrich and was diluted 1000-fold for Western blotting. His6-p53 and E6 protein of human papillomavirus type 16 were from Boston Biochem (Cambridge, MA, USA). Oligonucleotides were ordered from Integrated DNA Technologies (Coralville, IA, USA) with their sequences listed in Supplementary Table 1. Biotin-CoA was prepared by conjugating biotin-maleimide with Coenzyme A[17]. wt UB and xUB were expressed as fusions with an N-terminal ybbR tag and were labeled with biotin by the transfer of biotin-pantetheinyl group from biotin-CoA to the ybbR tag catalyzed by Sfp phosphopantetheinyl transferase[17, 65].

**Construction of the protein expression plasmids**. To construct human xUba1 mutant with six mutations (Q608R, S621R and D623R, E1037K, D1047K and E1049K), primers Bo184 and Bo185, and Bo 186 and Bo187 were paired to amplify Uba1 gene in pET-wt Uba1 by polymerase chain reaction (PCR). The amplified PCR fragments had mutations Q608R, S621R and D623R incorporated into the adenylation(A) domain of Uba1. The two PCR fragments were assembled by overlapping PCR and cloned into the pET-wt Uba1 vector between restriction sites FseI and EcoRI to generate pET-xUba1(A). To incorporate the three mutations in the UFD domain of Uba1, the mutated Uba1 gene in pET-xUba1(A) was PCR amplified with primers Bo13 and Bo73. PCR fragment was digested by restriction enzymes BamHI and EcoRI, and cloned into pET-xUba1(A) to generate pET-xUba1 with mutations in both the adenylation and the UFD domains. pET-UbcH7 with the R5E and K9E mutations was constructed by PCR amplifying the UbcH7 gene with primers WY9 and WY10 and cloned into pET28a between restriction sites NdeI and XhoI.

HECT domains of Sumrf1, Sumrf2, Nedd4, and E6AP with an N-terminal Flag tag were expressed with the pET28 vector. The genes of the HECT domains were amplified with primers WY1-8 by PCR. The amplified fragments were digested with restriction enzymes SacII and NotI, and cloned into the pET28a plasmid. To express the mutant E6AP HECT from yeast selection, the genes of the mutant HECT domains were PCR amplified with primers WY4 and WY8 from the corresponding pCTCON2 vector, digested by SacII and NotI, and cloned into pET28a. For the expression of full-length E6AP, PCR primers WY21 and WY8 were used to amplify the full-length gene from pGEX4-wt E6AP and cloned into the pET28a-Flag vector between restriction sites SacII and NotI.. For the expression of full-length xE6AP with mutated HECT domains, mutant HECT genes were amplified with primers WY4 and WY8 and cloned into the pET28-E6AP vector between restriction sites PstI and NotI. The CDK1 and CDK4 genes were PCR amplified from their mammalian cell expression plasmids with primer pairs K1-K2 and K3-K4, respectively. The PCR fragments were digested with NedI/XhoI and ScaI/NotI, respectively, and cloned into the pET-28a plasmid. The UbxD8 gene was amplified from a mammalian expression plasmid with primers LZ1 and LZ2, digested with NdeI and NotI, and cloned into pET28. The β-catenin gene was loned into pGEX plasmid. The pET or PGEX plasmids were transformed into BL21(DE3) pLysS chemical competent cells (Invitrogen) for protein expression.

**Construction of the E6AP library**. The gene of the E6AP HECT domain was PCR amplified from pET-E6AP with primers WY11 and WY12. The amplified fragment was double-digested with NheI and XhoI, and cloned into pCTCON2 plasmid to generate pCTCON2-E6AP HECT. To generate E6AP library in the pCTCON2 plasmid, the E6AP HECT domain gene was PCR amplified with WY13 and WY14 to incorporate randomized codons at residues 651, 652, 653, 654, and 656. The PCR fragment amplified with WY13 and WY14 was combined with fragment amplified with WY11 and WY12 to assemble the HECT library gene by overlapping extension with primers WY11 and WY12. The amplified library gene was digested with NheI and XhoI, and cloned into the pCTCON2 vector. Transformation of the plasmid library into XL1 blue electro-competent cells afforded a library of $2.0 \times 10^8$ in diversity, large enough to cover all the mutants with randomized residues replacing D651, D652, M653, M654 and T656 in the E6AP HECT domain. Transformed XL1 blue cells were plated on LB-ampicillin plates (LB agar supplemented with 100 μg mL$^{-1}$ ampicillin) and allowed to grow at 37 °C overnight. Colonies growing on the plate were scraped and the library DNA was extracted with the Plasmid Maxiprep Kit (Qiagen).

**Yeast display of the E6AP library**. The E6AP library in pCTCON2 was chemically transformed into EYB100 yeast cells[66, 67]. Briefly, yeast cells were first cultured at 30 °C in 200 ml YPD (20 g dextrose, 20 g peptone, and 10 g yeast extract in 1 L deionized water, sterilized by filtration) to an optical density at 600 nm (OD600) around 0.5. The cells were then pelleted at $1,000 \times g$ for 5 min. Cells were washed by 20 mL TE (100 mM Tris base, 10 mM EDTA, pH 8.0) and 20 mL LiOAc-TE (100 mM LiOAc in TE), before resuspension in approximately 800 μL LiOAc-TE. A typical transformation reaction contained a mixture of 1 μg pCTCON2 plasmid DNA, 2 μL denatured single-stranded carrier DNA from salmon testes (Sigma-Aldrich), 25 μL re-suspended yeast competent cells, and 300 μL polyethylene glycol (PEG) solution (40% (w/v) PEG 3350 in LiOAc-TE). To achieve a library size of $10^6$, 30 transformations were set up in parallel. Control was also prepared in which the pCTCON2 plasmid was excluded. Both the transformation reactions and the control were incubated at 30 °C for 1 h and then at 42 °C for 20 min. Cells in each transformation were pelleted by centrifuging at $1000 \times g$ for 30 s and re-suspended in 20 mL SDCAA medium (2% (w/v) dextrose, 6.7 g Difco yeast nitrogen base without amino acids, 5 g Bacto casamino acids, 50 mM sodium citrate, and 20 mM citric acid monohydrate in 1 L deionized water, sterilized by filtration). Yeast cells were re-suspended, pooled together into 1 L SDCAA medium, and allowed to grow at 30 °C over a 2-day period to an OD600 above 5. For long-term storage of the yeast library, 20 ml of the yeast culture was aliquoted in 15% glycerol stock and stored at −80 °C. To titer the transformation efficiency, 10 μL of the re-suspended yeast transformants was serially diluted in SDCAA medium and plated on Trp−plates (20 g agar, 20 g dextrose, 5 g (NH4)2SO4, 1.7 g Difco yeast nitrogen base without amino acids, 1.3 g drop-out mix excluding Trp in 1 L deionized water, and autoclaved). Yeast cells transformed with pCTCON2 plasmids would appear within 2 days of incubation at 30 °C.

**Model selection of yeast cells displaying E6AP HECT domain**. Yeast cell EYB100 was transformed with pCTCON2—wt E6AP HECT and streaked on a Trp−plate. After two days of incubation at 30 °C, cells were scraped from the Trp−plate to inoculate in a 5 mL SDCAA culture that was allowed to shake at 30 °C to reach an initial OD600 of 0.5. Cells were centrifuged at $1000 \times g$ for 5 min and induced for E6AP HECT expression by resuspension in 5 mL SGCAA (2% (w/v) galactose, 6.7 g Difco yeast nitrogen base without amino acids, 5 g Bacto casamino acids, 38 mM Na2HPO4 and 62 mM NaH2PO4, in 1 L deionized water, sterilized by filtration). The yeast culture was shaken at 20 °C for 16–24 h. For analysis of E6AP display on the surface of yeast cells, $10^6$ cells were re-suspended in 0.1 mL Tris-buffered saline (TBS) (25 mM Tris, pH 7.5, 150 mM NaCl) with 0.1% bovine serum albumin (BSA). The cells were first labeled with biotin-wt UB based on UB loading on the E6AP HECT domain displayed on the cell surface. 100 μL labeling reaction

was set up with 0.5 μM wt Uba1, 5 μM wt Ubch7, 0.1 μM biotin-wt UB in a buffer containing 10 mM $MgCl_2$ and 50 mM Tris-HCl (pH 7.5). The reaction was incubated for 2 h at 30 °C, and then mixed with 100 μL 3% BSA. A mouse anti-HA antibody (Santa Cruz Biotechnology, sc-7392) was added to the reaction mixture to detect the expression of E6AP tag by binding to the HA tag fused to the N-terminus of the HECT domain. The anti-HA antibody was added to a final concentration of 10 μg mL$^{-1}$ and the cells were incubated for overnight at 4 °C. The cells were then washed twice with 0.1% BSA in TBS and stained with 5 μg mL$^{-1}$ goat anti-mouse antibody conjugated with Alexa Fluor 647 (Life Technologies, A21235) in 0.1 mL 0.1% BSA in TBS. 5 μg mL$^{-1}$ streptavidin conjugated with PE (Life Technologies, S866) was also added to bind to biotin-UB conjugated to the HECT domain. The cell suspension was shielded from light and incubated at 4 °C for 1 h. After washing twice with 0.1% BSA in TBS, cells were analyzed on a flow cytometer (BD LSRFortessa) to count the number of cells that were labeled with fluorophore. Cells were also analyzed from control labeling reactions in which the primary anti-HA antibody was excluded from the labeling reaction, or Uba1, UbcH7 or biotin- wt UB was excluded from the UB loading reaction.

**Selection of the E6AP library displayed on the yeast cell.** The first round of selection of the yeast library was carried out with magnetic-activated cell sorting (MACS). For subsequent rounds of selection, fluorescence-activated cell sorting (FACS) was used. For MACS, 500 μL reactions in TBS buffer (10 mM $MgCl_2$, 50 mM Tris-HCl, pH 7.5) with 0.1% BSA were set up with approximately $5 \times 10^7$ yeast cells displaying the E6AP library. The reaction mixture contains 5 μM xUba1, 20 μM xUbcH7 and 5 μM biotin-xUB to enable xUB transfer to HECT. After reacting for 2 h at 30 °C, cells were pelleted by centrifugation and re-suspended in fresh 0.1% BSA in TBS. This procedure was repeated twice to remove biotin-xUB that was not covalently conjugated to yeast cells. After washing, cells were mixed with 100 μL streptavidin-coated microbeads provided by the mMACS Streptavidin Starting Kit (Miltenyi Biotec, 130-091-287) in a total volume of 1 mL TBS with 0.1% BSA. Cell suspension was shielded from light and incubated at 4 °C for 1 h. The suspension of the cells and magnetic beads were then added to 30 mL of 0.1% BSA in TBS. The cell suspension was pelleted by centrifugation at $500 \times g$ for 10 min. The supernatant was aspirated, and the cell pellet including the magnetic beads was re-suspended in 500 μL 0.1% BSA-TBS. Yeast cells bound to magnetic beads by biotin–streptavidin interaction were captured by a magnet according to manufacturer's instructions, and the beads were washed with 0.1% BSA in TBS. Cells bound to the magnetic beads were eluted into 5 ml SDCAA medium supplemented with 100 μg mL$^{-1}$ ampicillin, and 50 μg/mL kanamycin, and were cultured at 30 °C overnight. In parallel, library cells were bound to primary and secondary antibodies to evaluate the display of HECT mutants on the yeast cell surface.

For the 2nd round of selection, the library cells amplified from the first round were incubated with 1 μM xUba1, 10 μM xUbcH7, and 5 μM biotin-xUB for 1 h. After loading biotin-xUB to the HECT domain, the cells were labeled with 10 μg mL$^{-1}$ mouse anti-HA antibody for 1 h. Next, the cells were washed three times, each time with 1 mL 0.1% BSA in TBS. The cells were incubated with 5 μg mL$^{-1}$ goat anti-mouse antibody conjugated with Alexa Fluor 647 and 5 μg mL$^{-1}$ streptavidin conjugated with PE as secondary reagents. After incubation for 1 h at 4 °C, the cells were pelleted, washed twice each time with 1 mL 0.1% BSA in TBS. Cells doubly labeled with both PE and Alexa Fluor 647 fluorophores were collected by FACS (BD FACSAria IIu). The cells collected were pelleted and re-suspended in SDCAA supplemented with 100 μg mL$^{-1}$ ampicillin and 50 μg mL$^{-1}$ kanamycin. The cells were allowed to grow at 30 °C to an OD$_{600}$ between 1 and 2. Glycerol stock of the cells were prepared and they were used to inoculate yeast cell culture for the next round of selection.

In subsequent rounds of selection, the concentration of xUba1, xUbcH7 and biotin-UB in the reaction were decreased in each round. For the 6th rounds of selection, 0.5 μM xUba1, 5 μM xUbcH7 and 0.1 μM xUB was used. The gate for sorting the yeast cells also became more stringent in each round with the sixth round only collecting the top 0.5% of doubly labeled cells. After six rounds of cell selection, the collected cells were grown in an SDCCA medium to an OD$_{600}$ around 0.5. Zymoprep II Yeast Plasmid Miniprep Kit (Zymo Research, D2004) was used to extract the pCTCON2 plasmid DNA. The plasmid was transformed into XL1 blue competent cells. Plasmid DNA from individual colonies were miniprepped, and sequenced to reveal the mutations in the selected HECT domain clones.

**Construction of lentiviral vector and stable cell lines.** To generate pLenti6-V5-D-TOPO-Asc1-hygromycin-HBT-(x)UB plasmids, HBT tag was sub-cloned from pQCXIP HBT-UB and fused with DNA fragments of human wt UB or xUB by PCR. The assembled DNA fragment was cloned into the pLenti6 plasmid with a hygromycin resistant gene. Genes of xUba1, xUbcH7 and xE6AP were cloned into lentiviral vectors for the selection of stable cell lines. Flag-xUba1 gene was PCR amplified with primer WY15 and primer WY16 and cloned into pLenti6-V5-D-TOPO-Flag-Asc1-blasticidin vector between restriction sites EcoRI and AscI. V5-xUbcH7 gene was PCR amplified from pET-xUbcH7 with PCR primers WY17 and WY18, digested with restriction enzymes Afe1 and NheI, and cloned into pLenti4-V5-D backbone with a zeocin-resistance gene. The gene of xE6AP was PCR amplified with primers WY19 and WY20 and cloned into a pLenti-puromycin vector with a myc tag between restriction sites NheI and XhoI. Virus packaging,

virus infection and selection of stable cell lines were performed according to the manufacturer's protocol for the ViraPower Lentiviral Expression System. Stable HEK293 cell lines expressing Flag-xUba1 and V5-xUbcH7 were selected with 10 μg mL$^{-1}$ blasticidin and 100 μg mL$^{-1}$ Zeocin, respectively. Stable cell line for Myc-xE6AP was selected with 1 μg mL$^{-1}$ puromycin. Expression of transfected genes was induced by the addition of 1 μg/mL doxycycline to the medium.

**Tandem affinity purification of xUB-conjugated proteins.** Tandem purification of cellular proteins conjugated to HBT-xUB was performed as following[23]. 30 dishes (10 cm in diameter) of HEK293 cells stably expressing the xUba1-xUbch7-xE6AP cascade were acutely infected with lentivirus HBT-xUB for 72 h. To inhibit proteasome activity, cells were treated with 10 μM MG132 for 4 h at 37 °C. Cells were then washed twice with ice-cold $1 \times$ PBS, pH 7.4, and harvested by cell scraper with buffer A (8 M urea, 300 mM NaCl, 50 mM Tris, 50 mM $NaH_2PO_4$, 0.5% NP-40, 1 mM PMSF and 125 U/ml Benzonase, pH 8.0). For Ni-NTA purification, cell lysates were centrifuged at $15,000 \times g$ for 30 min at room temperature. 35 μL of Ni$^{2+}$ Sepharose beads (GE Healthcare) for each 1 mg of protein lysates were added to the clarified supernatant. After incubation overnight at room temperature in buffer A with 10 mM imidazole on a rocking platform, Ni$^{2+}$ Sepharose beads were pelleted by centrifugation at $100 \times g$ for 3 min and washed sequentially with 20-bead volume of buffer A (pH 8.0), buffer A (pH 6.3), and buffer A (pH 6.3) with 10 mM imidazole. After washing the beads, proteins were eluted twice with 5-bead volume of buffer B (8 M Urea, 200 mM NaCl, 50 mM $Na_2HPO_4$, 2% SDS, 10 mM EDTA, 100 mM imidazole, 250 mM imidazole, pH 4.3). For streptavidin purification, the pH of the eluted fractions were adjusted to pH 8.0. 50 μL streptavidin-sepharose beads (Thermo Scientific, Rockford, IL) was added to the elution to bind ubiquitinated proteins. After incubation on a rocking platform overnight at room temperature, streptavidin beads were pelleted and washed sequentially with 1.5 mL buffer C (8 M Urea, 200 mM NaCl, 2% SDS, 100 mM Tris, pH 8.0), buffer D (8 M Urea, 1.2 M NaCl, 0.2% SDS, 100 mM Tris, 10% EtOH, 10% Isopropanol, pH 8.0) and buffer E (8 M urea, 100 $NH_4HCO_3$, pH 8).

**Sample digestion.** Residual buffer E was removed and 200 μL of 50 mM $NH_4HCO_3$ was added to each sample, which were then reduced with dithiothreitol (final concentration 1 mM) for 30 min at 25 °C. This was followed by 30 min of alkylation with 5 mM iodoacetamide in the dark. The samples were then digested with 1 μg of lysyl endopeptidase (Wako) at room temperature for 2 h and further digested overnight with 1:50 (w/w) trypsin (Promega) at room temperature. Resulting peptides were acidified with 25 μL of 10% (v/v) formic acid (FA) and 1% (v/v) triflouroacetic acid (TFA) and desalted with a Sep-Pak C18 column (Waters). Briefly, the Sep-Pak column was washed with 1 mL of methanol and 1 mL of 50% (v/v) acetonitrile (ACN). Equilibration was performed with 2 rounds of 1 mL of 0.1% (v/v) TFA in water. The acidified peptides were then loaded and the column washed with 2 rounds of 1 mL 0.1% (v/v) TFA. Elution was carried out by 2 rounds of 50% (v/v) ACN (400 μL each) and the resulting peptide eluent dried under vacuum.

**LC-MS/MS analysis.** Liquid chromatography coupled to tandem mass spectrometry (LC-MS/MS) on an Orbitrap Fusion mass spectrometer (ThermoFisher Scientific, San Jose, CA) was performed at the Emory Integrated Proteomics Core (EIPC)[68, 69]. The dried samples were re-suspended in 10 μL of loading buffer (0.1% (v/v) formic acid, 0.03% (v/v) trifluoroacetic acid, 1% (v/v) acetonitrile), vortexed for 5 min and centrifuged down at maximum speed for 2 min. Peptide mixtures (2 μL) were loaded onto a 25 cm × 75 μm internal diameter fused silica column (New Objective, Woburn, MA) self-packed with 1.9 μm C18 resin (Dr. Maisch, Germany). Separation was carried out over a 2-hour gradient by a Dionex Ultimate 3000 RSLCnano system at a flowrate of 350 nL/min. The gradient ranged from 3 to 80% (v/v) buffer B (buffer A: 0.1% (v/v) formic acid in water, buffer B: 0.1% (v/v) formic acid in ACN). In each cycle, the mass spectrometer performed a full MS scan followed by as many tandem MS/MS scans allowed within the 3-second time window (top speed mode). Full MS scans were collected in profile mode at 120,000 resolution at m/z 200 with an automatic gain control (AGC) of 200,000 and a maximum ion injection time of 50 ms. The full mass range was set from 400–1600 m/z. Tandem MS/MS scans were collected in the ion trap after higher-energy collisional dissociation (HCD) in the ion routing multipole. The precursor ions were isolated with a 0.7 m/z window and fragmented with 30% collision energy. The product ions were collected with the AGC set for 10,000 and the maximum injection time set to 35 ms. Previously sequenced precursor ions within ±10 p.p.m. were excluded from sequencing for 20 s using the dynamic exclusion parameters and only precursors with charge states between 2+ and 6+ were allowed.

**Database search.** All raw data files were processed using the Proteome Discoverer 2.0 data analysis suite (Thermo Scientific, San Jose, CA). The database was downloaded from Uniprot and consists of 90,300 human target sequences. Peptide matches were restricted to fully tryptic cleavage and precursor mass tolerances of ±20 p.p.m. Dynamic modifications were set for methionine oxidation (+15.99492 Da), asaparagine and glutamine deamidation (+0.98402 Da), lysine ubiquitination (+114.04293 Da) and protein N-terminal acetylation (+42.03670). A maximum of 3 dynamic modifications were allowed per peptide and a static modification of

+57.021465 Da was set for carbamidomethyl cysteine. The Percolator node within Proteome Discoverer was used to filter the peptide spectral match (PSM) false discovery rate to 1%[70].

**Bioinformatics analysis**. Ingenuity Pathway Analysis (IPA) software (http://www.ingenuity.com) was used to map and identify the biological networks and molecular pathways with a significant proportion of genes having E6AP ubiquitination targets. Fisher exact test in Ingenuity Pathway Analysis software was used to calculate $p$-values for pathways and networks. The level of statistical significance was set at a $p$-value < 0.05. IPA was also used to visualize the identified biological networks. Proteins identified by the OUT screen were also analyzed by the CRAPome database (http://www.crapome.org/)[28]. CRAPome is based on data from interactome studies that carried out affinity purification under denaturing conditions. In contrast, the tandem purification for OUT screens was performed under more stringent denaturing conditions.

**Lentiviral silencing of E6AP**. Lentiviral GPIZ plasmids encoding shRNAs against E6AP (6 different shRNAs) were obtained from GE Dharmacon (Lafayette, CO), and lentiviruses were produced using the manufacturer's lentivirus packaging system and 293FT cells. HEK293 cells were infected with each lentivirus, followed by selection with puromycin for stable cell populations. The efficiency of gene silencing in each shRNA group was determined by immunoblotting using stable cell populations. For functional restoration, HEK293 cell population stably expressing anti-E6AP shRNA #1 was infected with the lentivirus packaged with pLenti6-Myc-wt E6AP.

**In vitro assay to confirm the substrates of E6AP**. All assays were set up in 30 μL TBS supplemented with 10 mM MgCl₂ and 1.5 mM ATP. In each UB transfer reaction, 5 μM of potential substrates (MAPK1, CDK1, CDK4, PRMT5, β-catenin, and UbxD8) were incubated with 1 μM wt Uba1, 5 μM wt UbcH7, 10 μM E6AP, and 20 μM wt UB for 2 h at 30 °C. The reactions were quenched by boiling in Laemmli buffer with BME, and analyzed by Western blotting probed with substrate-specific antibodies.

**Co-immunoprecipitation and to confirm E6AP substrates**. Transfection of pLenti-E6AP into the HEK293 cells was conducted with the Lipofectamine® 2000 according to the manufacturer's protocol. To immunoprecipitate substrate proteins, cells were treated with 10 μM MG132 (American Peptide, Sunnyvale, CA) for 90 min at 72-h post-transfection. HEK293 cells (80–90% confluent monolayer in 75 cm² cell culture flask) expressing control plasmid, shE6AP, shE6AP + E6AP cDNA, and E6AP cDNA were washed twice with ice-cold PBS, pH 7.4. 1 mL ice-cold RIPA buffer was added to cell monolayer and incubated with cell for at 4 °C for 10 min. The cells were disrupted by repeated aspiration through a 21-gauge needle. The cell lysate was transferred to a 1.5 mL tube. The cell debris was pelleted by centrifugation at 13,000 r.p.m. for 20 min at 4 °C and the supernatant was transferred to a 1.5 mL centrifuge tube and precleared by adding 1.0 μg of the appropriate control IgG (normal mouse or rabbit IgG corresponding to the host species of the primary antibody). 20 μL of re-suspended volume of Protein A/G PLUS-agarose was added to the supernatant and incubation was continued for 30 min at 4 °C. The agarose beads were pelleted by centrifugation at 350 × g for 5 min at 4 °C. From the cleared cell lysate, volume containing 2 mg total protein was transferred to a new tube. 30 μL (i.e., 6 μg) primary antibody was then added and incubation was continued for 1 h at 4 °C. After incubation, 50 μL of re-suspended volume of Protein A/G PLUS-Agarose was added. The tubes were capped and incubate at 4 °C on a rocking platform overnight. The agarose beads were pelleted by centrifugation at 350 × g for 5 min at 4 °C. The beads were then washed 4 times each time with 1.0 mL PBS. After the final wash, the beads were re-suspended in 40 μL 1 × Laemmli buffer with BME. The samples were boiled for 5 min and analyzed by SDS-PAGE and Western blot probed with antibodies specific for the substrate proteins.

**E6AP induced protein degradation**. To examine the effects of E6AP on steady-state levels of the substrates, HEK293 cells (5 × 10⁶ cells) were transiently transfected with 0.5, 1, 2, and 4 μg pLenti-wt E6AP with Lipofactamine 2000. Cells were harvested at 48-hour post-transfection and the amount of substrate proteins in the cell lysate was assayed by immunoblotting with substrate-specific antibodies. For cycloheximide (CHX) chase assays, HEK293 cells (5 × 10⁶ cells) were transiently transfected with 4 μg empty pLenti or pLenti-E6AP plasmids. After 48 h, cells were treated with 100 μg/mL CHX to block de novo protein synthesis and the cells were harvested after variable length of incubation time with CHX. The amount of substrate proteins in the cell were assayed by immunoblotting with antibodies against each substrate proteins. Protein levels were normalized to tubulin. Alternatively, CHX chase assays were performed on HEK293 cells stably expressing anti-E6AP shRNA to measure the effect of decreased expression of E6AP on substrate stability. Uncropped scans of the Western blots are presented in Supplementary Figs. 6–11.

**Data availability**. The proteomics data supporting the findings of this study have been deposited to the ProteomeXchange Consortium via the PRIDE partner repository with the identifier PXD005584 (http://proteomecentral.proteomexchange.org). All other data are available from the corresponding authors upon reasonable request.

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

## Acknowledgements

We thank Wade Harper, Jon M. Huibregtse, Linda Hicke, and K. Dane Wittrup for providing the gene constructs of Uba1, E6AP, Rsp5, and yeast display vector. This work is supported by grants provided from the National Institutes of Health (GM104498 to J. Y. and H.K.; CA112282 to H.K.), the National Science Foundation (1420193 and 1710460 to J.Y.), the Chicago Biomedical Consortium (Catalyst-026 to H.K. and J.Y., PDR-010 to X.L.), Project 985 startup grant (WF220417001 and WF114117001/004 to B. Z.), the Lynn Sage Breast Cancer Research Foundation, and the Department of Pharmacology at Northwestern University.

## Author contributions

J.Y. and H.K. conceived the idea, Y.W. conducted most experiments, X.L. developed the scheme for cell line construction, tandem affinity purification and target validation, L.Z. contributed to substrate verification, D.D. performed proteomic analysis, K.B., B.Z., and R.L. contributed to cloning and protein ubiquitination assays, H.Z. synthesized biotin-xUB, Y.B. performed bioinformatics analysis, J.Y. wrote the manuscript with Y.W., X.L. and H.K., and all authors approved the final version.

## Additional information

**Competing interests:** The authors declare no competing financial interests.

