## [Peer Review File · Nature Communications]

Reviewers' comments:

Reviewer #1 (Remarks to the Author):

The manuscript by Wang and colleagues is much improved compared to the previous version submitted to [XXXXX]. The more stringent criteria to generate a list of potential E6AP ubiquitination targets seems much more reasonable. The approach is clever and seems to work well. The general concept is interesting. I remain somewhat concerned about the validity of the proposed E6AP targets. Four potential, so far unidentified, substrates (MAPK1, PRMT5, CDK1, CDK4) are analyzed in detail to validate the list of E6AP substrates. Several different assays are employed, but most of them, except the shRNA knockdown followed by IP and anti-ubi blot (Fig. 6 lanes 2 and 3), are really asking whether E6AP can ubiquitinate these proteins and not if E6AP actually does ubiquitinate these substrates under physiological conditions. This is because all other assays use overexpression of E6AP. The experiment in Fig. 6 is a strong approach to answer the question of whether the test protein is a substrate. Closer inspection of the shRNA experiment (Fig. 6) does raise some questions about conclusions drawn from this experiment. First, the effect on Cdk4 ubiquitination is very subtle and not very convincing. (2) The interpretation of the result for PRMT5 is somewhat troubling because the major ubiquitin signal is at a molecular weight that is lower than that of unmodified PRMT5 (about 65kDa). The ubiquitin signal analyzed is thus likely not attached to PRMT5 (the signal should start at >65kDa). Note that in vitro ubiquitination of PRMT5 is also very inefficient with E6AP. Nevertheless, considering all validation experiments in combination there is some validity to the author's claim that these proteins are substrates. Overall this issue still bothers me though, because many readers will use the published table as strong evidence that this is a list of E6AP substrates. I do not believe this can be concluded based on the validation experiments, because they largely rely on E6AP overexpression and validation of 2 out of 4 candidates presented is questionable under physiological conditions. The manuscript is, however, quite interesting conceptually and many readers will be interested. If the authors could expand their validation by using a cycloheximide type degradation assay in combination with E6AP knockdown to provide more confidence in the generated substrate list, the overall paper will be very strong.

Reviewer #3 (Remarks to the Author):

The revised manuscript by Wang and colleagues entitled "Identifying the Substrate Proteins of E6AP by Orthogonal Ubiquitin Transfer" describes a novel strategy to identify E3-ligase substrates via engineering mutants of the ubiquitin conjugation pathway. Using in vitro studies and overexpression in a cell line, the authors demonstrate their method can be used to identify substrates of E6AP using enrichment of mutant Ub and LC-MS/MS. The authors added a third replicate of their proteomic screen and took the intersect of the three experiments, but seem to have lowered their criteria (PSM >2 instead of >3). I feel this cutoff has been chosen for pragmatic reasons (whatever could be validated) but it is important to know that their use of PSM instead of real quantification of MS intensities or stable isotope labeling is not considered vigorous in the proteomics field.

Nevertheless, the method is novel and of interest to the ubiquitin field and their addition of experiments to address some of my previous concerns is noted. It is probably true that the proteomics quantification is not going to dramatically change the results of their screen although I would imagine that it would do a lot to convince readers of the veracity of their results.

I think this manuscript is suitable for publication in Nature Communications.

Reviewer #4 (Remarks to the Author):

I agree with other reviewers that the OUT approach for E3 ligase substrate identification is clever and is shown here to be potentially useful. Issues remain, however, with the revised manuscript.

My first concern is that the approach is so complex that few other investigators are likely to use it. It is also not clear that it will be adaptable to other E3s, either HECT or non-HECT. Reasonable arguments notwithstanding, identifying another xHECT E3 would have gone a long way toward overcoming this criticism.

While the authors discuss isolated substrates that had been previously identified there is not an adequate discussion of previously identified substrates that were not isolated with their system. I am thinking particularly of the E6AP proteomics paper from the Howley and Harper labs. Furthermore, in the overall experimental design, I am extremely surprised that the authors did not undertake a comparison of cells that did or did not express the HPV E6 protein. This would have presumably resulted in isolation of several proteins, including p53, only in the E6-expressing cells, and perhaps a loss of normal cellular targets in the E6-expressing cells. This might have represented a very valuable internal validation of the approach, and they just might have learned something new, too. In the end, we are left with a small number of somewhat "validated" substrates of E6AP, but given the caveats of the validation (e.g., protein overexpression) pointed out in the first review, we are left with no new insights into the biological functions of E6AP in normal cells, HPV-positive cervical cancers, or neurological disease (Angelman syndrome or autism).

Finally, the written English remains poor in several areas. Particular examples are the first two sentences of the abstract and the first sentence of the introduction. Contrary to what was written in the response to reviewer 3, the first sentence of the introduction was apparently not corrected in review.

Reviewer #1 (Remarks to the Author):

The manuscript by Wang and colleagues is much improved compared to the previous version submitted to [XXXXX]. The more stringent criteria to generate a list of potential E6AP ubiquitination targets seems much more reasonable. The approach is clever and seems to work well. The general concept is interesting. I remain somewhat concerned about the validity of the proposed E6AP targets. Four potential, so far unidentified, substrates (MAPK1, PRMT5, CDK1, CDK4) are analyzed in detail to validate the list of E6AP substrates. Several different assays are employed, but most of them, except the shRNA knockdown followed by IP and anti-ubi blot (Fig. 6 lanes 2 and 3), are really asking whether E6AP can ubiquitinate these proteins and not if E6AP actually does ubiquitinated these substrates under physiological conditions. This is because all other assays use overexpression of E6AP.

Response: We followed the reviewer's suggestion (see below) and used cycloheximide (CHX) chase assay to compare the stability of the identified E6AP substrates in HEK293 cells with and without decreased expression of E6AP by shRNA (Fig. 7). We found the stabilities of PRMT1, CDK1, CDK4, β -catenin, and UbxD8 are enhanced in cells with decreased expression of E6AP compared to control cells without the expression of shRNA against E6AP. MAPK1, due to its high expression level, does not show significant changes of stability in cells with E6AP shRNA. The assay of CHX chase on cells expressing shRNA against E6AP avoided the ubiquitination of substrates due to overexpression of E6AP. Such data add another layer of evidence that the method of "Orthogonal Ubiquitin Transfer (OUT)" can identify the ubiquitination targets of E6AP in the cell.

The experiment in Fig. 6 is a strong approach to answer the question of whether the test protein is a substrate. Closer inspection of the shRNA experiment (Fig. 6) does raise some questions about conclusions drawn from this experiment. First, the effect on CdK4 ubiquitination is very subtle and not very convincing.

Response: We agree with the reviewer that the Western blot in Fig. 6e did not give a clear comparison of the ubiquitination levels of CDK4 in HEK293 cells, HEK293 expressing shRNA against E6AP, HEK293 expressing both E6AP and the E6AP shRNA, and HEK293 overexpressing E6AP. We have thus repeated immunoprecipitation of CDK4 from the four cell types, and used Western blotting to compare the ubiquitination levels of CDK4. Our new results showed a significant decrease in CDK4 ubiquitination in cells with inhibited expression of E6AP comparing to HEK293 cells or cells with overexpressed E6AP (Fig. 6e).

(2) The interpretation of the result for PRMT5 is somewhat troubling because the major ubiquitin signal is at a molecular weight that is lower than that of unmodified PRMT5 (about 65kDa). The ubiquitin signal analyzed is thus likely not attached to PRMT5 (the signal should start at >65kDa). Note that in vitro ubiquitination of PRMT5 is also very inefficient with E6AP.

Response: We repeated the *in vitro* ubiquitination of PRMT5 by E6AP and acquired better results (Fig. 5b). Still, PRMT5 expressed from *E coli* cells mainly showed mono-ubiquitinated product. The stability of PRMT5 in HEK293 cells is significantly enhanced with inhibited expression of E6AP, and is significantly decreased with overexpression of E6AP (Fig. 7d). These results should substantiate that PRMT5 is an E6AP target in the cell. The low molecular weight species below 65 kDa in Fig. 6c could be originated from proteolytic fragments of PRMT5 with UB attached.

Nevertheless, considering all validation experiments in combination there is some validity to the author's claim that these proteins are substrates. Overall this issue still bothers me though, because many readers will use the published table as strong evidence that this is a list of E6AP substrates. I do not believe this can be concluded based on the validation experiments, because they largely rely on E6AP overexpression and validation of 2 out of 4 candidates presented is questionable under physiological conditions. The manuscript is, however, quite interesting conceptually and many readers will be interested. If the authors could expand their validation by using a cycloheximide type degradation assay in combination with E6AP knockdown to provide more confidence in the generated substrate list, the overall paper will be very strong.

Response: We thank the reviewer's suggestion on performing CHX chase in combination with E6AP knockdown to verify the identified substrates. As stated above, we performed CHX chase to compare the stabilities of the identified substrates in HEK293 cells and in cells with inhibited expression of E6AP. We found PRMT5, CDK1, CDK4, β -catenin, and UbxD8 are stabilized due to the inhibition of E6AP expression. Among these substrates, β -catenin and UbxD8 are the newly verified E6AP substrates from the list of the OUT screen (Supplementary Table 1). The results of their *in vitro* ubiquitination, changes in ubiquitination level upon inhibition of E6AP expression, and protein stabilities correlated with E6AP expression are shown in Fig 5, 6 and 7, respectively.

Reviewer #3 (Remarks to the Author):

The revised manuscript by Wang and colleagues entitled "Identifying the Substrate Proteins of E6AP by Orthogonal Ubiquitin Transfer" describes a novel strategy to identify E3-ligase substrates via engineering mutants of the ubiquitin conjugation pathway. Using in vitro studies and overexpression in a cell line, the authors demonstrate their method can be used to identify substrates of E6AP using enrichment of mutant Ub and LC-MS/MS. The authors added a third replicate of their proteomic screen and took the intersect of the three experiments, but seem to have lowered their criteria (PSM >2 instead of >3). I feel this cutoff has been chosen for pragmatic reasons (whatever could be validated) but it is important to know that their use of PSM instead of real quantification of MS intensities or stable isotope labeling is not considered vigorous in the proteomics field.

Nevertheless, the method is novel and of interest to the ubiquitin field and their addition of experiments to address some of my previous concerns is noted. It is probably true that the proteomics quantification is not going to dramatically change the results of their screen although I would imagine that it would do a lot to convince readers of the veracity of their results.

I think this manuscript is suitable for publication in Nature Communications.

Response: We thank the reviewer's suggestions on assigning E6AP substrates based on "real quantification of MS intensities or stable isotope labeling". We certainly wish to incorporate these techniques for E3 substrate identification in future screens with OUT.

Reviewer #4 (Remarks to the Author):

I agree with other reviewers that the OUT approach for E3 ligase substrate identification is clever and is shown here to be potentially useful. Issues remain, however, with the revised manuscript.

My first concern is that the approach is so complex that few other investigators are likely to use it. It is also not clear that it will be adaptable to other E3s, either HECT or non-HECT. Reasonable arguments notwithstanding, identifying another xHECT E3 would have gone a long way toward overcoming this criticism.

Response: The reviewer's concern is legitimate, and we discussed the pros and cons of OUT at the end of the manuscript. We believe the yeast cell surface display method for engineering the HECT domain of E6AP could be used on other HECT E3s to connect them to the OUT cascade. The sequence and structural homology of various HECT domains may allow the mutations to be transplanted from the HECT domain of xE6AP to other HECT E3s to expand the reach of OUT. This would save time on engineering xE2-xE3 pairs from scratch. Furthermore, once an OUT cascade is developed for a E3 such as E6AP, it could be used by others to identify the substrates of a specific E3 in various cellular and physiological contexts. We agree it might take some effort to develop OUT cascades with various E3s. However, once it is available, the OUT cascade of an E3 would be a useful tool to study the biological functions of the E3 enzyme.

While the authors discuss isolated substrates that had been previously identified there is not an adequate discussion of previously identified substrates that were not isolated with their system. I am thinking particularly of the E6AP proteomics paper from the Howley and Harper labs.

Response: The paper of Howley and Harper (*Identification and proteomic analysis of distinct UBE3A/E6AP protein complexes, Mol Cell Biol, 2012, 32, 3095-3106*) used E6AP as a bait to screen for E6AP interacting proteins. It identified proteins that are involved in a complex with E6AP including HERC2, NEURL4, and MAPK6. It did not report whether E6AP targeted these proteins for ubiquitination. We do not see these proteins in Supplementary Table 1 listing the potential substrates of E6AP identified by OUT.

Furthermore, in the overall experimental design, I am extremely surprised that the authors did not undertake a comparison of cells that did or did not express the HPV E6 protein. This would have presumably resulted in isolation of several proteins, including p53, only in the E6-expressing cells, and

perhaps a loss of normal cellular targets in the E6-expressing cells. This might have represented a very valuable internal validation of the approach, and they just might have learned something new, too.

Response: We thank the reviewer's suggestion to use OUT to identify E6AP substrates in the presence of E6. This project is currently ongoing. Indeed, it would be of great interest in using OUT to characterize how E6 stirs the substrate profile of E6AP. We hope we will report our results soon.

In the end, we are left with a small number of somewhat "validated" substrates of E6AP, but given the caveats of the validation (e.g., protein overexpression) pointed out in the first review, we are left with no new insights into the biological functions of E6AP in normal cells, HPV-positive cervical cancers, or neurological disease (Angelman syndrome or autism).

Response: The significance of the current manuscript is to establish the feasibility of OUT in identifying E3 substrates. We hope we could soon follow up on various leads provided by the substrate profile of E6AP to investigate its role in cell regulation and diseases.

Finally, the written English remains poor in several areas. Particular examples are the first two sentences of the abstract and the first sentence of the introduction. Contrary to what was written in the response to reviewer 3, the first sentence of the introduction was apparently not corrected in review.

Response: We apologize for the poor English. We have revised the sentences flagged by the reviewer.

Reviewers' comments:

Reviewer #1 (Remarks to the Author):

The additional experiments support some of the previous results. I therefore support publication of the manuscript Nature Communications.

Reviewer #4 (Remarks to the Author):

The revised manuscript is marginally improved. I agree with the other reviewers that the OUT approach is novel and will be of interest to the field, however the validation of substrates is less than I would like to see, with no biological effects being correlated with ubiquitination of any of the newly identified substrates. While the authors imply that such studies will be part of the follow-up, the fact remains that everything from title on down is focused on "identifying the ubiquitination targets of E6AP". Some specific issues that remain are detailed below.

The supplemental data does not indicate proteins that were identified only in the OUT control (minus-xE6AP). That is, we see proteins that were identified in both the minus- and plus-xE6AP samples, but not with only the minus-xE6AP control. Surely there must have been some in this category?

In the validation experiments it would have been good to see some positive controls. That is, none of the "accepted" E6AP substrates (such as RAD23a and RNF2) were examined in the experiments shown in validation experiments shown in Figures 5-7. The most troublesome feature of the validation experiments is that in Figure 6, with shE6AP expression, there is simply no change in the steady state levels of any of the new substrates. While the authors responded enthusiastically to my previous suggestion of looking at HPV E6-expressing cells (to take advantage of internal positive controls, including p53) they have chosen to not do this at this time. This is unfortunate as this would give us a better indication of whether we can believe that the newly identified substrates are bona fide.

Response to the reviewers' comments on manuscript NCOMMS-16-29888A

Reviewer #1 (Remarks to the Author):

The additional experiments support some of the previous results. I therefore support publication of the manuscript Nature Communications.

Response: We appreciate the reviewer's critiques that have guided us to significantly improve our work.

Reviewer #4 (Remarks to the Author):

The revised manuscript is marginally improved. I agree with the other reviewers that the OUT approach is novel and will be of interest to the field, however the validation of substrates is less than I would like to see, with no biological effects being correlated with ubiquitination of any of the newly identified substrates. While the authors imply that such studies will be part of the follow-up, the fact remains that everything from title on down is focused on "identifying the ubiquitination targets of E6AP". Some specific issues that remain are detailed below.

The supplemental data does not indicate proteins that were identified only in the OUT control (minus-xE6AP). That is, we see proteins that were identified in both the minus- and plus-xE6AP samples, but not with only the minus-xE6AP control. Surely there must have been some in this category?

Response: We generated Supplementary Table 5 that lists 35 proteins that were consistently identified in the "minus-xE6AP control" cells that expressed HBT-xUB and the xUba1-xUbcH7 cascade without xE6AP. These proteins were not identified in the cells expressing the full OUT cascade of xUba1-xUbxH7-xE6AP.

In the main text (page 9), we added: "Also, 35 proteins were identified in multiple biological replicates of control cells without expression of xE6AP (Supplementary Table 5). Those proteins were not present in the samples purified from the cells expressing the full OUT cascade of E6AP."

In the validation experiments it would have been good to see some positive controls. That is, none of the "accepted" E6AP substrates (such as RAD23a and RNF2) were examined in the experiments shown in validation experiments shown in Figures 5-7.

Response: We followed the reviewer's suggestion to use HHR23A (RAD23a) as a positive control in the ubiquitination and degradation assays (Figs. 5g, 6h, and 7). As previously reported in references 32, 34 and 37, HHR23A is a E6AP substrate independent of E6. Indeed, we observed ubiquitination of HHR23A by E6AP *in vitro* and in HEK293 cells (Fig. 5g and 6h), the acceleration of HHR23A degradation upon E6AP overexpression (Fig. 7a and 7b), and the stabilization of HHR23A upon inhibition of E6AP expression by shRNA (Fig. 7c and 7d). These results validate our ubiquitination assays on the E6AP substrates identified by OUT.

The most troublesome feature of the validation experiments is that in Figure 6, with shE6AP expression, there is simply no change in the steady state levels of any of the new substrates.

Response: For the assay of ubiquitination in Figure 6, we used the proteasome inhibitor MG132, which we think was why there were essentially no change in the steady-state levels of the substrates. The legend of Fig. 6 stated: “After 1.5-hour treatment of cells with MG132, ubiquitination of each target protein was compared among the blank HEK293 cell (HEK293), HEK293 expressing shE6AP (shE6AP), HEK293 expressing both shE6AP and recombinant E6AP (shE6AP+OE), and HEK293 expressing recombinant E6AP (OE).” The use of MG132 was intended to block substrate degradation, so that the extents of ubiquitination could be compared between cells treated with the shRNA of E6AP and cells with overexpression of E6AP. As expected, we found that the levels of HHR23A, a previously confirmed E6AP substrate, were comparable among shE6AP, OE and control cells after MG132 treatment (Fig. 6h).

We measured the change of the steady-state levels of the substrates in Fig. 7a. There we increased E6AP expression in the cell without pretreatment with MG132. We did observe the decreases in the steady-state levels of the substrates MAPK1, PRMT5, CDK1, CDK4, UbxD8 and the positive control HHR23A with increasing E6AP expression.

While the authors responded enthusiastically to my previous suggestion of looking at HPV E6-expressing cells (to take advantage of internal positive controls, including p53) they have chosen to not do this at this time. This is unfortunate as this would give us a better indication of whether we can believe that the newly identified substrates are bona fide.

We appreciate the reviewer’s suggestion to use OUT to differentiate E6AP substrates that are dependent or independent on E6. While this part of work is ongoing in our laboratories, it remains preliminary, mostly due to the complex nature of the biological system. For example, the requirement for E6 in E6AP-mediated ubiquitination of particular substrates in cells does not seem to be black and white simple. Consequently, validation of each candidate for E6-dependent and independent substrates is far more time-consuming. We are trying to carefully define the E6 dependency using the OUT platform and subsequent biological validations. In our opinion, the complexity and time-consuming nature apparently makes it difficult to include the study into the current manuscript.

The scope of the current manuscript is to establish OUT as a feasible platform to identify E6AP substrates. According to the excellent advice from this reviewer, we now demonstrate that HHR23A, a known E6AP substrate independent of E6, is among the E6AP targets identified by OUT (Supplementary Table 1) and reconfirm its ubiquitination by E6AP *in vitro* and in the cell (Fig. 5g and 6h), and its degradation induced by E6AP expression (Fig. 7). Thus, HHR23A serves as an “internal positive control” to validate our experimental strategy and establish the feasibility of OUT in identifying E6AP substrates.